# Microcomb-based integrated photonic processing unit

Bowen Bai[1,7], Qipeng Yang[1,7], Haowen Shu[1,7], Lin Chang ®[1,2,3,7] ✉, Fenghe Yang[4], Bitao Shen[1], Zihan Tao[1], Jing Wang[5], Shaofu Xu[5], Weiqiang Xie[2], Weiwen Zou ®[5], Weiwei Hu[1], John E. Bowers ®[2] ✉ & Xingjun Wang ®[1,3,6] ✉

The emergence of parallel convolution-operation technology has substantially powered the complexity and functionality of optical neural networks (ONN) by harnessing the dimension of optical wavelength. However, this advanced architecture faces remarkable challenges in high-level integration and on-chip operation. In this work, convolution based on time-wavelength plane stretching approach is implemented on a microcomb-driven chip-based photonic processing unit (PPU). To support the operation of this processing unit, we develop a dedicated control and operation protocol, leading to a record high weight precision of 9 bits. Moreover, the compact architecture and high data loading speed enable a preeminent photonic-core compute density of over 1 trillion of operations per second per square millimeter (TOPS mm$^{-2}$). Two proof-of-concept experiments are demonstrated, including image edge detection and handwritten digit recognition, showing comparable processing capability compared to that of a digital computer. Due to the advanced performance and the great scalability, this parallel photonic processing unit can potentially revolutionize sophisticated artificial intelligence tasks including autonomous driving, video action recognition and image reconstruction.

Artificial intelligence (AI) with deep learning[1] has witnessed remarkable success in data-heavy computational tasks. Driven by the great demand of computing speed and energy efficiency from AI, optical neural networks (ONNs) have gone through rapid progress in the last decade[2–4]. Benefiting from the low-loss, large-bandwidth and high-coherency nature, leveraging photons for conducting many fundamental operations in neural networks, including Fourier transform[5], convolution[6,7], and matrix multiplication[8,9], can significantly boost the computing speed and lower the energy consumption. Particularly in the recent years, the successful implementations of ONN in photonic integrated circuits (PICs) have opened the possibility of manufacturing photonic-based computational chips[10–12] using existing semiconductor manufacturing infrastructures. Due to the compactness, scalability, and energy efficiency, such novel photonic processing units (PPUs), as a contrast to the electronic-based graphic processing units (GPUs)[13], potentially can revolutionize the hardware for AI.

Inspired by the general algorithm for designing unitary matrix[14], Shen et al.[8] demonstrated a photonic processing unit using a Mach-Zehnder interferometer (MZI) mesh to accelerate matrix computations, driven by a single wavelength laser. Following this route, rapid progression has been achieved within a short time. Now, these processing units have been significantly extended in scale and successfully commercialized. However, the large footprint of the MZI mesh and the clock synchronization problem of multiple high-speed input signals

[1]State Key Laboratory of Advanced Optical Communications System and Networks, School of Electronics, Peking University, Beijing 100871, China. [2]Department of Electrical and Computer Engineering, University of California, Santa Barbara, CA 93106, USA. [3]Frontiers Science Center for Nano-optoelectronics, Peking University, Beijing 100871, China. [4]Zhangjiang Laboratory, Shanghai 201210, China. [5]State Key Laboratory of Advanced Optical Communications System and Networks, Department of Electronic Engineering, Shanghai Jiao Tong University, Shanghai 200240, China. [6]Peking University Yangtze Delta Institute of Optoelectronics, Nantong 226010, China. [7]These authors contributed equally: Bowen Bai, Qipeng Yang, Haowen Shu, Lin Chang. ✉e-mail: linchang@pku.edu.cn; bowers@ece.ucsb.edu; xjwang@pku.edu.cn

make such single-frequency approaches[9,15] difficult to improve the compute density[16] of the photonic-core defined as the computing speed (TOPS) normalized by the photonic core area that conduct linear operations (mm²). On the other hand, to further exploit the potential of optical computing, advanced ONN demands high-level parallel information processing to boost the computational throughout, for which the wavelength division multiplexing (WDM) approach comes into play. Recently, the architecture of such a system has been successfully realized for photonic convolution operation[6], by harnessing the Kerr microcombs[17–19] that provide multiple equidistant optical frequency lines. These advances lead to significantly improved compute density, up to 0.2 TOPS mm⁻², with a greatly relieved modulation speed requirement of the modulator.

However, despite the appealing properties of WDM parallel ONN[7], this advanced architecture faces remarkable challenges in high-level integration: first, the microcomb generation in previous demonstrations all had to rely on bench-top lasers, since a chip-laser-based microcomb pumping scheme is quite challenging; second, in the information loading and processing units, the previous configuration was not compatible with the standard foundry-based silicon photonic platforms: they have to rely on either bulk fiber equipment or special fabrication process; finally, WDM operation in PICs requires much more complicated calibration and control procedures compared to the single wavelength approach. As a result, a WDM-based optical computing system with high-level integration so far remains elusive, which prevents the transition of parallel ONN from research lab to industry deployment.

In this work, we propose and demonstrate a PPU with all the essential photonic components integrated, including the multi-wavelength source, data loading session, and the data processing core. An AlGaAs-on-insulator dark-pulse microcomb provides a coherent multi-optical-channel source, which is directly pumped by an on-chip laser. A novel configuration of the convolution accelerator (can also be used as a reconfigurable microwave photonic filter[20]) has been proposed and all the processing components of it are integrated together: high-speed data flow is encoded on every comb tooth of the microcomb via on-chip silicon electro-optical modulator (EOM); kernel weights are mapped to voltages applied to the on-ring heaters in the microring resonator (MRR) array (which we term MRR weight bank in the following) and data caching is implemented by the embedded optical delay lines. Importantly, we developed a procedure for the calibration of this system, which enables the accurate control of all the individual components, and can be extended to the WDM system. Owing to the high integration level, this PPU exhibits a preeminent photonic-core (overall) compute density of 1.04 TOPS mm⁻² (0.104 TOPS mm⁻²), which is 5 times higher than the previous record[6] in WDM ONN architectures. Two proof-of-concept experiments for a convolutional neural network are performed, including image edge detection and digit recognition. The quality of edge detection and the accuracy (96.6%) of recognition are comparable to that of a digital computer. Our approach represents an essential step towards a fully integrated photonic processing unit for real-world deployment.

## Results
### Integrated parallel photonic processing unit based on microcombs

A fully integrated photonic hardware to perform convolution operation is conceptually illustrated in Fig. 1a. A DFB-pumped microcomb source serves as optical carrier. The combs are regrouped to map the convolution kernel matrices using an on-chip wavelength division demultiplexer. Input data matrices are converted to RF domain by a signal generator. The input RF signals are amplified by a driver and broadcast onto comb lines via the high-speed on-chip silicon EOM.

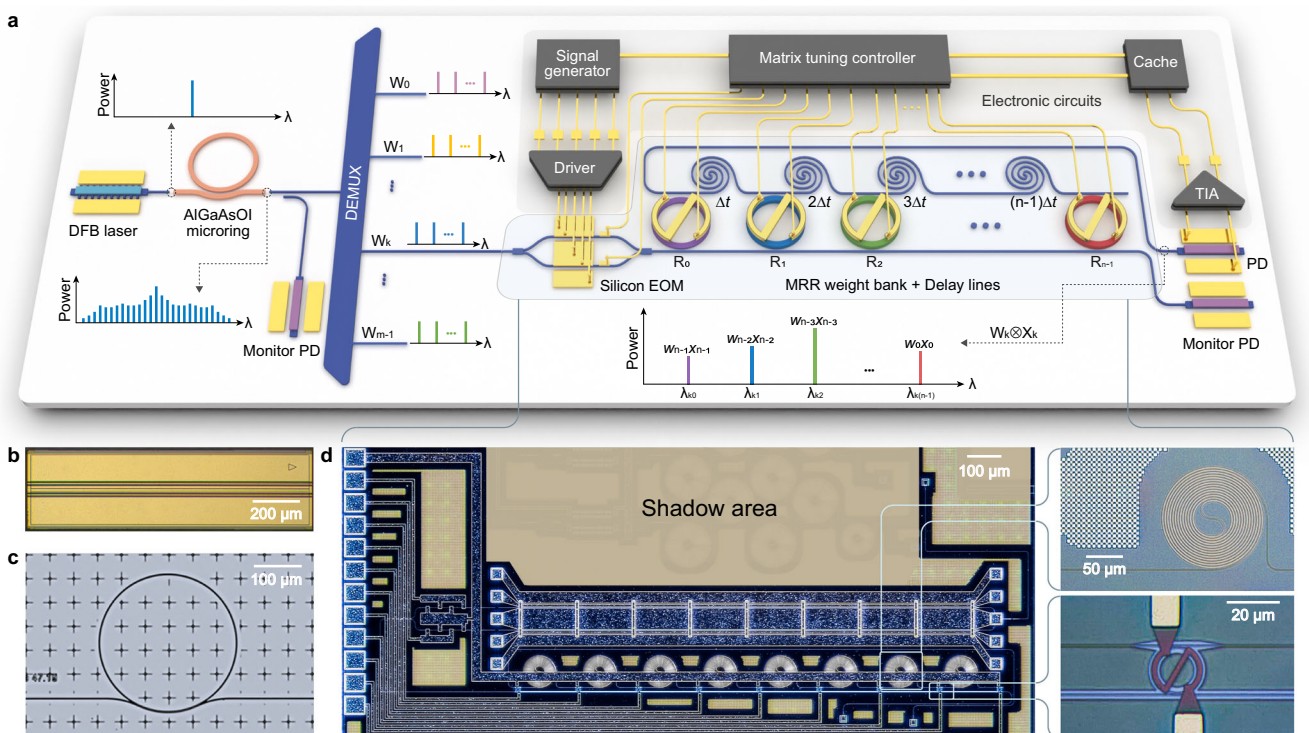

Fig. 1 | Integrated photonic processing unit (PPU). a Conceptual drawing of the fully integrated PPU. b Optical image of the commercial tunable DFB laser chip for microcomb generation. c Top-view image of the AlGaAsOI microresonator. The radius of the microresonator is 144 μm, corresponding to a free spectral range (FSR) of 91 GHz. d Optical micrograph of a fabricated photonic chip. The silicon electro-optic modulator (EOM) is monolithically integrated with the microring resonator (MRR) weight bank. The zoom-in micrographs on the right illustrate the on-chip Si spiral waveguide delay line (top) and MRR with in-ring heater (bottom). DEMUX demultiplexer, TIA trans-impedance amplifier, PD photodetector.

The matrix tuning controller transforms the kernel weights to voltages applied to the on-ring heaters and the thermal phase shifters in the silicon EOM. The add-drop MRR weight bank performs spectrum slicing, kernel weight loading, and spectrum recombination simultaneously. Si spiral waveguides are embedded to introduce on-chip time delays. The photodetector (PD) at the thru-port monitors the resonance states of the MRRs to ensure the stability of the weight loading. By sampling the RF output signals (generated from the PD at the drop port and amplified by the TIA) at appropriate moments, the computation results can be obtained. When calibrating the PPU, the matrix tuning controller receives data and instructions from the outside world and loads the processed input data to the signal generator. The convolution results are stored in the cache for conducting the system-level calibration procedure.

The integrated microcomb used in our work (see in Fig. 1c) is a microresonator fabricated on the AlGaAsOI platform[21]. The fabrication of this device is based on a wafer-bonding process, which can be adopted by current commercial heterogeneous III-V/Si photonic foundries[22]. The comb is operated at dark-pulse state originated from mode-crossing between fundamental and high-order mode. Thanks to the high third-order nonlinear coefficient of AlGaAs ($n_2 = 2.6 \times 10^{-17}$ m$^2$ W$^{-1}$), this coherent state can be accessed under pump power at a few milliwatt-level under a moderate Q factor[23]. Meanwhile, the dark-pule microcomb has high conversion efficiency and large access window in the spectrum, which is pumped by an off-the-shelf commercial InP DFB laser chip (see in Fig. 1b). Compared to another approach we previously demonstrated for microcombs directly pumped by integrated laser[24–26], the AlGaAsOI dark pulse microcombs is more power efficient and with larger reconfigurability[21,27]. Importantly, it does not require additional chip-to-chip or heterogeneous integration, which is more flexible to implement in existing photonic infrastructures.

The silicon photonic integrate circuit (Fig. 1d) with the same structure reported in our latest work[20] is the core of the photonic processing unit, consisting of data-loading session and computing core. It is fabricated by a commercial SiPh foundry in a one-to-one 200 mm SOI wafer run with standard 90 nm lithography process. For input vector loading, a balanced Mach-Zehnder travelling-wave PN depletion modulator is implemented with two thermal phase shifters. The MRR weight bank, consisting of parallel-coupled add-drop MRRs, are capable of weighting the comb lines individually over a continuous range. To precisely control the kernel weights, on-ring TiN micro heaters are implemented to tune the resonances of the MRRs. To introduce true time delay, Si spiral waveguides with adiabatic Euler bends are used. The average time delay between adjacent comb lines is 58.88 ps (for the detailed time delay measurement, see Supplementary Note 4), corresponding to ~17 GBaud modulation rate.

### Operation for on-chip WDM-based convolution system

One major challenge for realizing chip-based WDM analog computation, compared to the bench-top system, is significantly higher sensitivity of the system to fabrication and environment variation that usually come from the optical power fluctuation, inner crosstalk, restricted linear dynamic range, etc. Therefore, a dedicated stabilization and control method has to be used for enabling the accuracy of the computing results. This critical problem has been proposed in previous papers[11,28], and here for the first time, we introduce a systematic protocol to overcome this challenge.

First, the fluctuation of the comb source can to be minimized by using dark-pulse microcombs in the AlGaAsOI platform. A typical spectrum of this microcomb (DFB pumped) is shown in Fig. 2a. Thanks to the relatively large thermo-optical effect of the AlGaAs and the inherent thermal stability of the dark pulse approach, high stability of the microcomb operation can be enabled without any feedback or control electronics[23]. As shown in Fig. 2b, the power deviations of the central 20 channels of the microcomb in Fig. 2a within 60 minutes are

recorded, and statistical analyzed. All of the comb channels maintain a power fluctuation of lower than 2 dB, which ensures the consistency and duration of the computing operation.

A more critical control challenge comes from the SiPh part. MRRs are sensitive to fabrication and thermal variations, so that effective calibration methods[29,30] have to be applied to precisely control the multi-channel MRR weight banks. To match the scale of the $2 \times 2$ kernel matrix, we selected four consecutive MRRs for calibration. A high-precision voltage scanning procedure was developed and implemented to automatically obtain the "weight-voltage" lookup table (Fig. 2c) of each MRR. The minimum voltage sweep step is 0.01 V, which is sufficient for coarse thermal tuning of the MRR. As the voltage increases, the variation of the weight shows a trend similar to the Lorentzian peak shape of the MRR and is distributed in the 0-1 interval.

In addition to the coarse calibration using lookup table before conducting convolution, we also applied an in-situ gradient-descent control (GDC) method (see "Methods") to compensate the fabrication imperfectness and cross-talk of the MRR weight bank (Supplementary Note 2). The effectiveness of the GDC method with a command weight matrix [0, 0.5; 0.5, 1] is shown in Fig. 2d. The accuracy of the weight loading is greatly improved with root mean square error (RMSE) dropping from 0.043 to $0.4 \times 10^{-3}$. To evaluate the GDC method for an arbitrary weight matrix, we measured the weights as a function of calculated weights for a given channel (Fig. 2f). The other channels are manipulated with a weight list of all combination of (0, 0.33, 0.66, 1) (see Supplementary Note 3 for details). The measured weights are tightly concentrated at the diagonal line (the target weights), which indicates negligible inter-channel cross-talk after calibration. The attached histograms show the deviation of the measured weights. Within three epochs, The maximum deviation for all weights is lower than $1 \times 10^{-3}$, which corresponds to a record-high weight control precision of 9 bits. The detailed calculation about the weight control precision is shown in Supplementary Note 3.

Furthermore, since the silicon EOM should operate in its linear dependence on the intensity of the microcomb lines, the calibration of the silicon EOM is also carried out in this work, including two aspects: (1) uniformity response to different wavelengths and (2) selection of appropriate operating point and driving voltage. In our design, we adopt a balanced SiP traveling-wave Mach-Zehnder modulator (TWMZM) under push-pull configuration to eliminate the effect of the interference spectrum (of the MZ architecture) on the flatness of the comb lines. For multi-level pulse amplitude modulation (PAM), the height of the eye diagrams and the evenness of the eye openings directly reflect whether the modulator is working in its linear region. Here, we choose four-level PAM for calibration. To achieve the best linearity and quality of the eye diagrams, the silicon EOM operates at the quadrature point with positive slope via carefully tuning the voltages applied to the thermal phase shifters ($V_{ps} = (2.5, 0)$ V) and the DC bias of the two traveling-wave electrodes ($V_{bias} = (2.332, 1.697)$ V). It should be noted that due to the cosine wave form of the MZM transfer function[31], the driving voltage should be limited to the linear part of the transfer function. Therefore, the peak-to-peak voltage ($V_{pp}$) of the differential RF input is kept below 300 mV. Figure 2e illustrates the PAM-4 eye diagrams after calibration. The 4 channels are opened independently (the weight is "1" for a given channel while the weight of the other three channel is "0") for the measurement. The height of the four eye diagrams is the same and the amplitude levels are almost equidistant.

The value of the kernel matrix is distributed within the range of $(-1, +1)$ after normalization, however, the proposed MRR weight bank cannot directly calculate negative values since the light intensity is always positive. To implement convolution in complete real-number domain, the original kernel matrix $W$ is decomposed into two matrices $W'$ and $W''$ as Fig. 2g shows. The linear transformation $w' = \frac{1}{2}(w+1)$

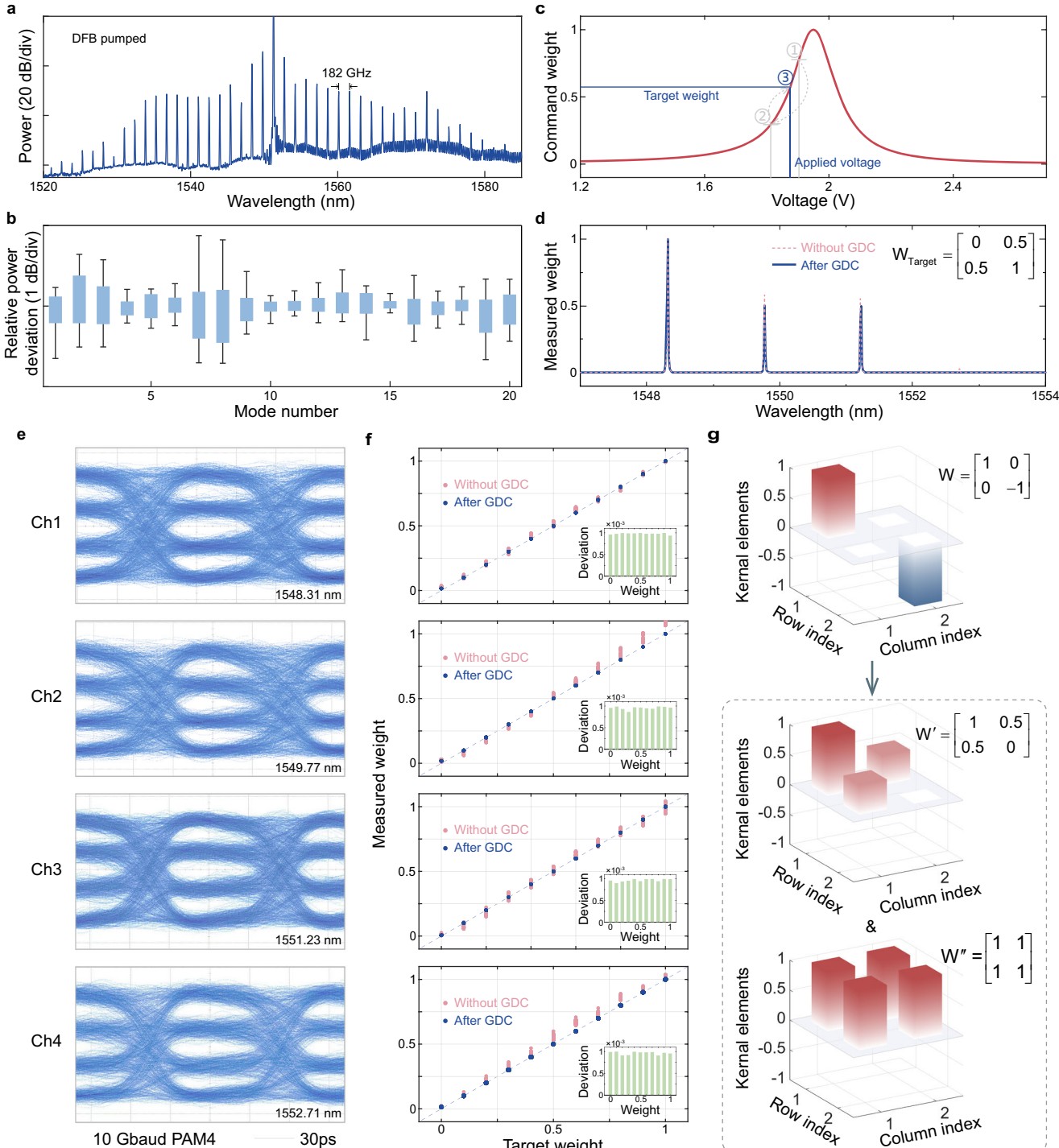

**Fig. 2 | Operation for real number WDM-based convolution system. a** Measured optical spectra of microcombs with 2-FSR (182 GHz) spacing. **b** Power deviations of the microcomb. **c** Normalized kernel weights vs. the voltage applied to the on-ring heater for one MRR. **d** The normalized spectrum of the microcomb lines when a specific weight matrix [0, 0.5; 0.5, 1] is implemented before (red dashed line) and after GDC (blue solid line). **e** PAM-4 eye diagrams (measured at drop port) of the 4 channels at 10 Gbaud symbol rate. **f** The measured weights as a function of calculated weights for a given channel when command weights are loaded on the other three channels. The inserts illustrate the deviation. **g** Kernel matrix decomposition for real-number convolution.

moves the elements in $W$ to non-negative space, and then, the convolution of $W$ and data matrix $X$ is written as

$$W \otimes X = (2W' - W'') \otimes X = 2W' \otimes X - W'' \otimes X, \quad (1)$$

where $W'$ is a matrix with all elements equal to 1, $X$ is the data matrix in which the elements are normalized gray-scale values (non-negative) of

the pictures. For example, the weight matrix [1, 0; 0, −1] (one kernel for image edge detection) can be decomposed into [1, 0.5; 0.5, 0] and [1, 1; 1, 1]. By conducting $W' \otimes X$ and $W'' \otimes X$ with same data matrix successively, truly real-valued convolution is achieved. Although negative-valued kernel has to be decomposed into two non-negative matrices, by sending the outputs from two MRR weight banks to the opposite ports of a balanced PD, real-valued convolution can be

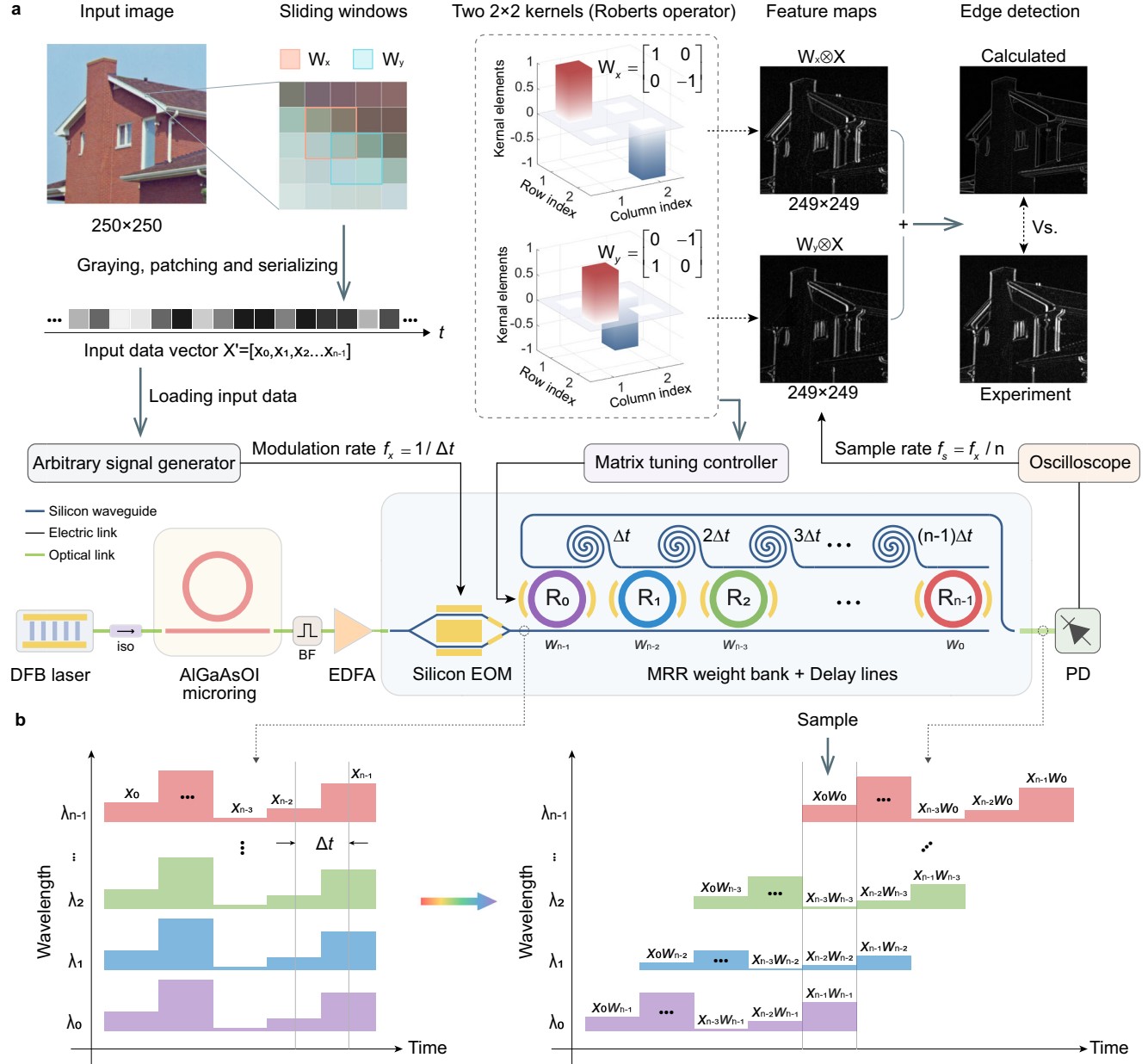

**Fig. 3 | Convolution for image processing. a** Schematic of photonic convolution and the experimental result of image edge detection with two 2 × 2 kernels (Roberts operator). A picture (house) from USC-SIPI image database is used as the input image. **b** Working principle of the microcomb-driven photonic processing unit. The wavelength-time plane is stretched by the on-chip optical delay lines with time delay step Δt. The colors of the MRRs and the time series represent the resonant wavelengths of the microrings. iso isolator, BF band-pass filter.

accomplished all at once. Such framework has been proposed in our recent work[20] and is under investigation.

## Convolution for image edge detection

The convolution of data matrix X and kernel matrix W is written as

$$W \otimes X = \sum_{i=0}^{n-1} w_i x_i = w_0 x_0 + w_1 x_1 + \cdots + w_{n-1} x_{n-1}, \quad (2)$$

where $n$ is size of the two matrices. The operation can be mapped into a plane of wavelength and time as illustrated in Fig. 3b. The gray-scale value of the pixels in the sliding window is normalized and serialized as an input vector $X'$. The element in vector $X'$ is encoded to the intensity of the $n$ parallel comb lines simultaneously via the silicon EOM. Then $n$ replicas of vector $X'$ flow to the calibrated MRR weight

bank with $n$ kernel weights (represent using different colours). The time delay between adjacent channels is $\Delta t = 58.88$ ps, which determines the modulation rate ($f_x = 1/\Delta t \approx 17$ Gbaud) and stretches the wavelength-time plane. At a appropriate time, the intensity of the overlapped comb lines is the result of the convolution for a given sliding window. A high-speed PD is used to receive the optical signals and naturally acts as a photonic adder. Note that the duration of the time slice is $(2n - 1)\Delta t$, the sample rate should be $f_s = f_x/n$. In our design, the kernel matrix size $n = 4$ and footprint (S) of the photonic computing core (four channels in use) is about $0.82 \times 0.16 \approx 0.131$ mm². Therefore, the photonic-core compute density of our proposed PPU is $2nf_x/S = 2 \times 4 \times 17 \times 10^9/0.131 \approx 1.04$ TOPS mm⁻². If the footprints of DFB pump laser, comb source, silicon weight bank chip and photo-detector are all taken into consideration. The total chip area is about 1.311 mm² and the corresponding overall

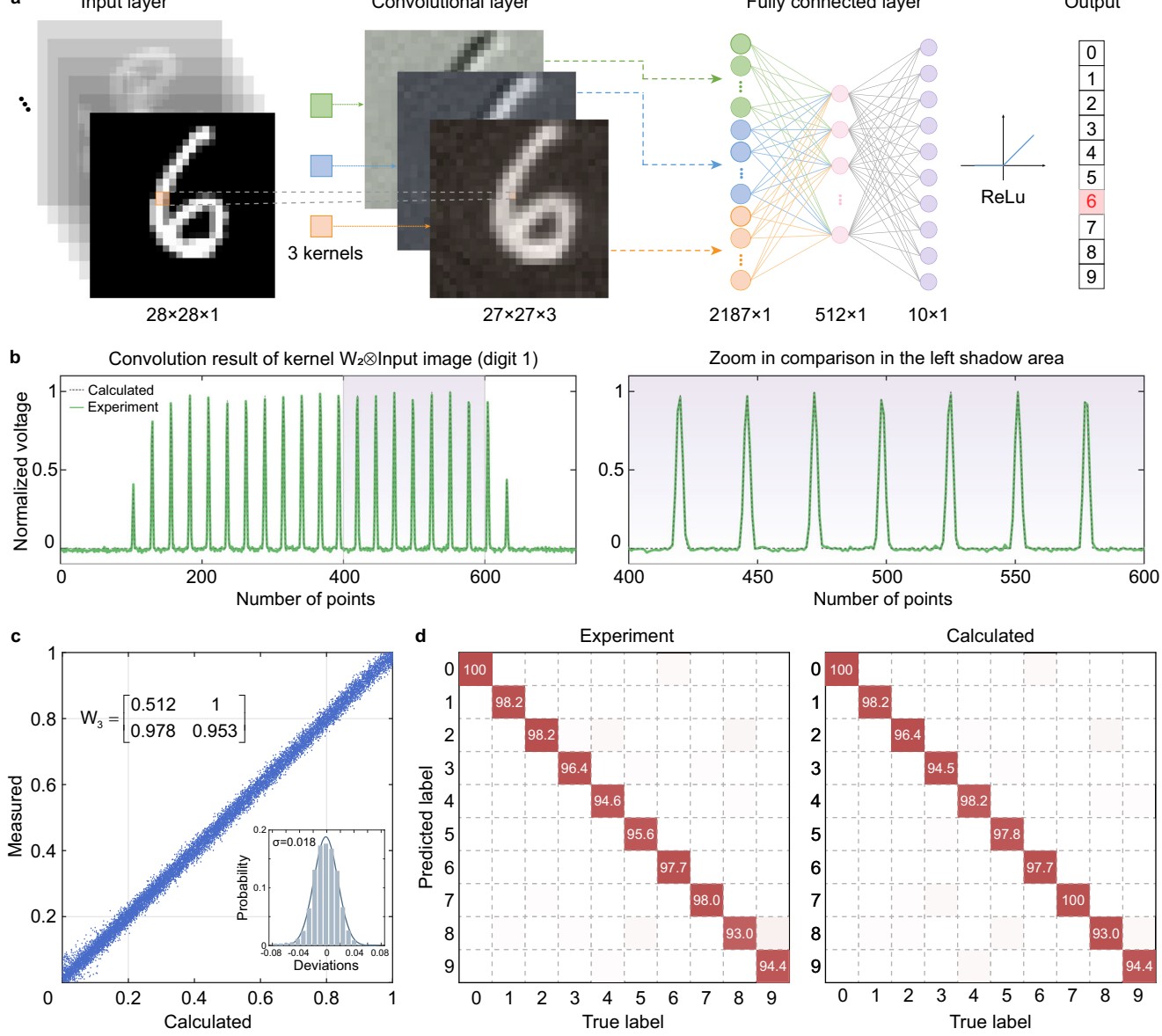

**Fig. 4 | Photonic convolutional layer in a CNN. a** Architecture of the convolution neural network (CNN) with three convolution kernel matrices ($W_1$ = [0.351, −0.729; −1, −0.094], $W_2$ = [0.551, 0.891; 0.761,1] and $W_3$ = [0.512,1; 0.978, 0.953]) for hand-writing digit recognition. The corresponding feature maps of image digit 1 are also illustrated. **b** The output RF signal after sampling for digit 1. The gray dashed line and green solid line are the calculated (ideal) and experimentally obtained waveform, respectively. **c** Scatter plot for convolution accuracy measurement with a fixed kernel matrix. The inset is a residual error distribution histogram showing a standard deviation of 0.018. **d** The confusion matrices of digit classification. The accuracy of the prediction results for experiment (96.6%) and calculation (97.0%) shows excellent agreement.

compute density is about 0.104 TOPS mm⁻² (see Supplementary note 8 for details).

To demonstrate the aforesaid principle experimentally, we conduct image edge detection (Fig. 3a) using two small-scale kernel matrices. The input image is firstly converted to a 250 × 250 gray pixel matrix with 8-bit values. Pixels in the sliding window are serialized into a 1 × 4 input vector $X'$ and loaded to the silicon EOM via an arbitrary waveform generator (AWG). The modulated comb lines flow to the MRR weight bank (controlled and calibrated by the matrix tuning controller) to conduct the convolutional operation. The kernels chosen for edges detection are the Roberts cross operator[32], which performs an approximation of the 2-D spatial gradient measurement on the image. Kernel $W_x$ highlights the −45° edges of the of the input image while $W_y$ extracts the opposite +45° edges. Finally, the optical convolution results then convert to RF outputs by an off-chip fast PD

and sampled using a high-speed real-time oscilloscope. The two feature maps after convolution are then combined to yield the experiment result, which highlights the sharp edges in the input image and agrees well with the calculated (ideal) one using a digital computer. This experiment verifies the feasibility and veracity of our proposed PPU to implement convolutional operations. For detailed description about the convolution procedure, see Supplementary Note 5.

### Handwritten digits classification

To further highlight the capability of the PPU demonstrated in this work, a handwritten digits classification task is conducted. The structure of the CNN we used here (shown in Fig. 4a) consists of one convolutional layer and two fully-connected layers (perform high-level reasoning). The input image is a matrix with a size of 28 × 28. The convolution layer first convolutes the input matrix with three kernels

and the results are then treated with nonlinear activation function (usually ReLu). As the kernels slides along the input matrix, the convolution operations yield three feature maps, which contribute to the input of the next fully-connected layers after serializing into a $2187 \times 1$ vector. The linear part of convolutional layers are decomposed to dot products by reusing the PPU temporally while the fully-connected layer is implemented on a computer due to the two massive weight matrices ($512 \times 2187$ and $10 \times 512$). In future works, the fully-connected layers could also be implemented on the chip if applying low-rank tensor decomposition[33] to the large weight matrices. The output layer with ReLu function gives the final classification results. The parameters in the CNN are trained with a standard back-propagation algorithm (see the Supplementary Note 6 for details) and the kernel weights are copied to the MRR weight bank via a matrix tuning controller. The input image is mapped to the RF signals in time series and encoded to the comb lines by silicon EOM at a baud rate of 17 GBaud. The input data convolves with three $2 \times 2$ kernels ($W_1$, $W_2$, and $W_3$) to generate three $27 \times 27$ feature maps.

The waveform obtained from the PD after sampling and the zoom in comparison between the experiment and calculated results are shown in Fig. 4b. The slight difference between the experiment and calculation is mainly caused by the imperfect linearity of the silicon EOM and the small random drift of the whole system during operation. To further inspect the operation accuracy of the convolution layer, we randomly choose 13,000 input values to convolve with a fixed $2 \times 2$ kernel matrix ($W_3$). The comparison between calculated and measured results are shown in Fig. 4c. The scatter points are closely distributed along the diagonal line (theoretical results), which indicates a small standard deviation of 0.018.

We then feed 500 images from MNIST handwritten digit dataset[34] into the PPU to test the accuracy of the classification. The confusion matrix (Fig. 4d) illustrates the accuracy of the prediction obtained from experiment and theoretical calculation is 96.6% and 97.0%, respectively. Details on the experimental setup and results are given in Supplementary Note 7. The conformable results indicate that the cross talk and noise has limited impact on the PPU performance after calibration. The structure of the CNN we used consists of one convolutional layer with 3 kernels and two fully-connected layers. The sampling rate is 17/4 GHz and the original kernel matrix $W$ is decomposed into two matrices $W'$ and $W''$. Since $W''$ is a matrix with all elements equal to 1 for all inputs, the duration to yield 3 feature maps is $27 \times 27 \times 4/(17\text{HZ}) \times 4 = 686.1$ ns. The run time of the digital backend (directly given by our computer: Intel(R) Core(TM) i7-10700K CPU) to implement the fully-connected layers is about 5390.6 ns. Therefore, the overall classification time is 6076.7 ns. Therefore, the PPU can process handwritten digit images at the speed of $1/(6076.7\,\text{ns}) = 1.65 \times 10^5\,\text{s}^{-1}$.

## Discussion

Thanks to the high modulation rate and the associated calibration procedure, the photonic-core compute density in this work is more than 5 times higher compared to previous WDM ONN, and a record high weight control precision of 9 bits is achieved. A comprehensive comparison with other integrated photonic computing architectures is provided in Supplementary Note 8. The performance of the PPU can be further improved by employing superior architecture design and optimized photonic devices. Since the photonic-core compute density of our architecture scales with the modulation rate and matrix size while is inversely proportional to the footprint (mainly occupied by the delay lines), if higher modulation rate were adopted, the footprint would drastically decrease due to the reduced length of the delay line. The kernel matrix size can be extended to $3 \times 3$ or $5 \times 5$ by utilizing more comb lines and a redesigned MRR weight bank. Although the relatively narrow bandwidth of the MRRs in our prototypical weight bank architecture will cause distortions and losses at high data loading

rate, by utilizing high order MRRs[35] combined with MZI-embedded microring[36], both the high-speed data loading and the weight tuning can be realized by the flat-top pass band and embedded MZI. Therefore, the photonic-core compute density of the PPU can reach 15 TOPS mm$^{-2}$ with the optimized structure at 50 Gbaud modulation rate. Considering $3 \times 3$ kernel size is sufficient for the needs of image processing in a convolutional layer, the number of current comb lines (see Fig. 2a) can support a parallel convolution with 5 kernels and further extended with broader spectrum.

As an analog computing system, the optical noise is one major limitation to the convolution accuracy of our PPU. Phase noise in the comb lines and erbium-doped fibre amplifier (EDFA) introduced amplified spontaneous emission (ASE) noise result in computational errors and this effect is more pronounced when the modulation rate scales up. The relatively high noise floor of the DFB laser and EDFA are the dominant factors to lower the optical signal to noise ratio (OSNR). By introducing a narrow bandwidth filter after the pump and an on-chip MRR for comb distillation[37], the OSNR can be significantly improved. By decomposing the original kernel into two kernels, the problem of loading negative weights can solved. More importantly, if the kernel weights are very close to 0, the original weight value 0 can be shifted to 0.5, avoiding the significant decrease of the OSNR due to the small output power.

The eternal pursuit for photonic computing is processing data at high speeds with low power consumption. In our proof-of-concept configuration, the power consumption mainly derives from the InP DFB pump laser, EDFA, silicon photonic chip, modulator drivers, TEC and digital backend. The power consumption estimated according to the components used in our measurement setup is about 89.5 W, of which about 90% of the energy consumption comes from the bench-top instruments (for details of the power consumption estimation, see Supplementary note 9). However, as a prototype, the energy efficiency of the PPU can be greatly improved. Superior integration techniques[38,39] and optimized photonic devices can be employed in pursuit of a fully integrated microcomb-driven PPU. Self-injection locked dark-pulse microcomb sources[26,40] can be monolithically realized utilizing heterogeneously integrated III-V lasers[25] and MRRs. The resonance states can be held with low-loss phase change materials[41] and a near "zero power consuming" MRR weight bank could be achieved. On-chip semiconductor optical amplifiers (SOA)[42] or circuit-based erbium-doped amplifier[43] can replace the discrete EDFA and be integrated before the PD to provide power compensation and non-linear activation[44]. The discrete PD can be replaced by integrated high-speed germanium-on-silicon PD[45]. All these processes are compatible with current III-V/SOI photonic foundries. Besides, the digital circuits, such as control unit, driver, TIA, etc. can be monolithically integrated with photonic devices[46], which further improves the compactness and power-efficiency of the PPU.

To show the ultimate potentials of our architecture, the excepted power consumption (estimated in line with the similar protocols in refs. 6,7) is about 1.05 W, corresponding to an excepted energy efficiency of 2.38 TOPS W$^{-1}$ for a fully integrated PPU with 50 Gbaud modulation rate and $5 \times 5$ kernel size. In the excepted energy efficiency calculation, the power consumption of the pump laser, the on-chip SOA (take the place of EDFA), and the electronic blocks (including integrated driver, TIA, digital-to-analog converter (DAC), analog-to-digital converter (ADC)) is included. The power consumption of the discrete EDFA, the digital backend and the temperature controllers is excluded. The detailed estimation procedure about the energy efficiency is shown in Supplementary Note 9. Although the temperature control and some effects in packaging (loading effect, coupling and interference effects, etc.) will cause extra power consumption, the potential energy efficiency advantage of the PPU is still significant.

In conclusion, we have demonstrated a microcomb-driven PPU with all the essential components integrated on chip. An associated

calibration procedure is developed to precisely operate the system, enabling a preeminent compute density capable of exceeding dozens of TOPS mm$^{-2}$. The performance of image edge detection and hand-written digit recognition is comparable to that of a digital computer, which portends that the proposed PPU is capable of processing more sophisticated AI tasks. Our work paves a way to a fully integrated photonic computing system and can potentially change the entire field of AI.

## Methods

### Design and fabrication of the devices

The AlGaAsOI microcomb used in our work was designed to work within the normal dispersion range at C band (waveguide dimensions: 400 nm height and 1000 nm width). The fabrication of AlGaAs resonators were based on heterogeneous wafer bonding technology using a 248 nm deep-ultraviolet (DUV) stepper for lithography. An optimal photoresist reflow process and dry etch process were developed to reduce the waveguide scattering loss. More fabrication details can be found in our previous work[21].

The silicon photonic circuit (the same one used in our recent work[20]), including high-speed silicon EOM and MRR weight bank, was designed on a 200 mm SOI wafer with a silicon-layer thickness of 220 nm and a BOX layer thickness 3 $\mu$m. The high-speed data loading unit is a balanced silicon Mach-Zehnder EOM with 2-mm-long travelling-wave electrode and two TiN thermal phase shifters. The weight bank consists of nine MRRs in an add/drop configuration and four sequential MRRs are in use as a proof-of-concept. The MRR weight bank is designed based on rib waveguides. The rib width is 450 nm and the height of the slab is 90 nm. A slight difference is introduced in the ring radii to match the 182 GHz wide (2 FSR) channel spacing (Fig. 2a) and the radii of the four MRRs are 8.8 $\mu$m, 8.813 $\mu$m, 8.826 $\mu$m, and 8.839 $\mu$m, respectively. The minimum gap between the ring and the bus waveguide is 240 nm. On-ring TiN heaters are implemented to precise control the kernel weights. All these silicon photonic devices are fabricated using 90 nm CMOS-compatible processes at CompoundTek Pte Ltd.

### Characterization of unit devices

The AlGaAsOI micro-resonator has a waveguide to lensed fibre coupling loss of 3–4 dB/facet. The ring waveguide width is 1 $\mu$m, which supports normal dispersion within the C band. The average Q factor of the AlGaAsOI MRR is around 2 million, corresponding to a waveguide loss of lower than 0.3 dB cm$^{-1}$. The coupling loss of the focused TE mode grating couplers is 4–5 dB. The typical OE 3-dB bandwidth of the depletion mode silicon EOM is > 25 GHz (Supplementary Note 1), measured by a vector network analyzer (Keysight N524). The insertion loss of the silicon EOM is about 4 dB and the phase shifters in the modulator are TiN micro heaters with resistance ~200 $\Omega$. Microring filters used for WDM and weighting are tuned by on-ring TiN micro heaters with resistance ~ 400 $\Omega$. A 182 GHz wide (2 FSR) channel selecting range can be obtained under 10 mW power dissipation. The TiN layer is about 1.2 $\mu$m above the silicon layer, ensuring a trade-off between the heating efficiency and absorption loss of the metal. The average time delay of the 2-$\mu$m wide silicon delay lines in the MRR weight bank is 58.88 ps with delay time variation of <3% and insertion loss < 0.5 dB among the 4 channels in use. More details about the time delay measurement can be found in Supplementary Note 4. The photodetector is a discrete 40 GHz bandwidth PD (Finisar HPDV2120R), which can be replaced by an on-chip high-speed waveguide type Ge/Si PD[47].

### Weight calibration with in-situ GDC

Due to the high sensitivity, precise MRR control is always a challenge, especially in a large scale MRRs system. The fabrication deviations and temperature drifts severely impact the weighting accuracy of the MRR

weight bank. Although each MRR can be calibrated by measuring the transmission-voltage lookup table, the convolution results may still deviate from the desired ones when all MRRs work simultaneously due to thermal crosstalk. To suppress the thermal crosstalk, we develop a calibration procedure using in-situ GDC method. The kernel weights are pre-trained on a computer and accurately mapped to the voltages applied to the on-ring heaters using the GDC method.

For a given channel in the MRR weight bank, assume the measured weight is $w^{(j)}$ and the target weight is $\hat{w}^{(j)}$. Since the 'weights-voltages' curve (Fig. 2c) can be pre-obtained using our automatic voltage scanning procedure, the $w^{(j)}$ can be expressed as

$$w^{(j)} = f(v^{(j)}). \tag{3}$$

Then, the loss function ($L$) is defined as the mean square error (MSR) of the weights

$$L^{(j)} = \frac{1}{2}\left(w^{(j)} - \hat{w}^{(j)}\right)^2 = \frac{1}{2}\left[f(v^{(j)}) - \hat{w}^{(j)}\right]^2, j = 0,1,\ldots,n-1 \tag{4}$$

where $j$ is the number of the given channel, $n$ is the total number of the channels in use, $v^{(j)}$ is the voltage applied to the on-ring heater. To minimize the loss function, the gradient descent method is employed and the derivatives of $L^{(j)}$ are formulated as

$$\frac{dL^{(j)}}{dv^{(j)}} = \left(w^{(j)} - \hat{w}^{(j)}\right)\frac{dw^{(j)}}{dv^{(j)}}. \tag{5}$$

If choosing the change of the weights in the opposite direction of the gradient of the $L^{(j)}$, the updated applied voltage can be written as

$$\begin{cases} v_{i+1}^{(j)} = v_i^{(j)} - \eta\frac{dL^{(j)}}{dv_i^{(j)}}, i = 0,1,2,\ldots,M^{(j)}-1 \\ |f(v_{M^{(j)}}^{(j)}) - \hat{w}^{(j)}| < \varepsilon, j = 0,1,2,\ldots,n-1 \end{cases} \tag{6}$$

where $\eta$ is a strictly positive hyper-parameter called the learning rate and the $M^{(j)}$ is the number of iterations. The calibration procedure does not stop until the absolute value of $(w^{(j)} - \hat{w}^{(j)})$ for all channels is less than a small number $\varepsilon$ ($\varepsilon = 0.001$ in our case).

Note that our calibration procedure not only eliminates the effects of thermal crosstalk in the MRR weight bank, but the weight interdependence caused by common parasitic resistance can be attenuated as well.

### Details for experimental setup

In the two proof-of-concept experiments, the dark-pulse microcomb is generated in a direct DFB-pumping manner. The DFB pump laser is packaged with a thermoelectric cooler (TEC) module (Phononic FBM-013865). An optical isolator is employed to avoid light reflection. A dark-pulse Kerr comb with 2-FSR spacing is generated with proper environmental temperature and pump frequency detuning around ~1551.3 from the blue side. The free-running comb is first amplified by a low-noise EDFA (Amonics AEDFA-PA-35-B-FA) and four comb lines are selected by an optical band-pass filter (EXFO XTM-50) before launching into the PPU. The input and output optical coupling are realized using focused grating couplers for the TE mode. The voltages applied to the phase shifters in the silicon EOM and the TiN micro-heaters on the MRR are tuned by four programmable direct-current power supplies (Keysight E36312A).

The measurement environment is shown in Supplementary Fig. S1a. The prototype PPU is packaged on a printed circuit board (PCB) with a hole in the center and supported by a 0.5-mm thick copper plate for higher thermal conductivity. The PCB is attached to a TEC with temperature control accuracy of 0.01 ℃ to reduce the thermal crosstalk.

The output light from the PPU is split by a 10:90 fiber coupler: 10% of the optical power is sent into an optical spectrum analyzer (Yokogawa AQ6370C) for kernel weight monitoring, while the other 90% of the light propagates to a discrete fast PD (Finisar HPDV2120R). A 50 GSa s$^{-1}$ arbitrary waveform generator (AWG, Tektronix AWG70001B) is employed to produce the differential input RF signals. The RF signals from the AWG are then amplified by two linear electrical drivers (SHF S807C) before routing to the silicon EOM. The output RF signals from the PD are received by a real-time oscilloscope (Agilent DSA-X 96204Q) with 80 GSa s$^{-1}$.

## Data availability

The data that supports the plots within this paper and other findings of this study are available on Figshare (https://doi.org/10.6084/m9.figshare.21687854). All other data used in this study is available from the corresponding authors upon reasonable request.

## Code availability

The codes that support the findings of this study are available from the corresponding authors upon reasonable request.

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

## Acknowledgements

The authors thank Jun Qin, Ming Jin, and Yuansheng Tao for helpful suggestions on the experiments and useful discussions on the manuscript, Shenzhen PhotonX Technology Co., Ltd, for laser packaging support. The authors also appreciate the support from Yan Zhou and Zhangfeng Ge during the electrical packaging. The nano-fabrication facility of UCSB was used.

## Author contributions

X.W. initiated this project and B.B. conceived the research and methods. The silicon photonic chip was designed by B.B., H.S., and J.W. The experiments were conceived and conducted by B.B., Q.Y., H.S. and S.X. The data processing and modelling is conducted by Q.Y. The AlGaAsOI microresonator structure is conceived and prepared by L.C., W.X., and H.S. The Kerr comb generation strategy is developed by H.S. and L.C. The electrical packaging is accomplished by F.Y and Z.T. Other characterizations are conducted by B.B. and Q.Y, with the assistance from H.S., B.S., and S.X. The results are analyzed by Q.Y. and B.B. All authors participated the discussion of the research. The project was coordinated by B.B under the supervision of W.Z., W.H., L.C., J.E.B., and X.W.

## Competing interests

The authors declare no competing interests.
