## [Peer review file · Nature Communications]

REVIEWER COMMENTS

Reviewer #1 (Remarks to the Author):

In this manuscript, the authors present two experiments using a photonic processing system that utilizes a microcomb as the multi-wavelength source, achieving a 9-bit weight resolution. The proposed scheme is successfully used for edge detection and hand-written digit recognition.

Here are my comments:

1. Novelty - While this work shows an interesting engineering demonstration, the novelty should be explained more clearly. Multi-wavelength photonic computing systems have been demonstrated before. Integrated Kerr combs as multi-wavelength sources (e.g. for multi-wavelength communication) have been demonstrated before. In the abstract, the claim that “both the multi-wavelength source and the fast processing core are integrated” is not accurate. They are separately implemented and coupled through an isolator and an EDFA. Please elaborate on the novelty. I suggest adding a sentence on convolution implementation method (as a novelty aspect) to the abstract.

2. Performance - The characterization of performance is somewhat inaccurate. The authors indicate 1 Tops/mm² is achieved. However, a true metric is classification time. In the implemented system, a single linear photonic computation layer is implemented, which could improve the overall compute time, but the bottleneck could be the digital processing (backend). What is the true overall compute/classification time? Does the reported $5.83 \times 10^6/s$ include the digital backend? What is the equivalent Tops/mm² for overall compute/classification? Is the chip area of electronic control blocks, the microcomb and the pump laser included in the compute density calculations? In the abstract, the claim that the system can be used for “video action recognition” should be quantitatively explained. For video classification a deep network is often needed. Given that the implementation is a single linear photonic layer, and digital backend is needed, is this system suitable for video classification? Please provide quantitative justification.

3. Power consumption – The power consumption calculations should be modified:

a. First, the power consumption should be calculated considering the wall-plug efficiency (WPE) of the pump laser and the power consumption of the pump temperature controller. From Supp. material, it seems that the on-chip optical power of the pump is used for power consumption calculations which is not accurate. I suggest re-calculating the power consumption considering the WPE of the pump laser and the power consumption of the temperature control system.

b. The power consumption of the EDFA (shown in Fig. 3) should be considered as it is part of the system.

c. The power consumption of modulator drivers used for MZM drive should be considered.

d. It seems a fast sampling scope is used as a part of the detection system. What is the plan to replace the scope for a practical use? What would be the power consumption overhead? Similarly, programmable power supplies are used.

e. It is inaccurate to consider the power consumption of different electronic blocks based on best reported numbers in the literature (for TIA, ADC, DAC, ...) for the reported system. First, these blocks are implemented on different processes (14nm CMOS FinFET, 7nm CMOS, ...) and there is no guarantee that they will yield the same results if implemented on the same process. Second, once connected together (either monolithically or through hybrid packaging), the loading effect, the noise effect, the coupling and interference effects most likely require re-design of these blocks and most likely results in substantial increase in the power consumption. Third, these blocks are not part of the measurement systems. The power consumption of the actual components used in the measurement setup should be considered. Also, what is the power consumption of the digital backend? It should be included in the calculations as well.

4. Scalability – There are multiple aspects to the scalability of this work:

a. Microcomb: the design is very nice. For scalability to a larger number of lines ($n > 4$), corresponding to a larger kernel size, it is not clear that a larger number of lines at the same per-line power could be generated. What is the effect of unequal power of comb lines on the classification accuracy and system power consumption? Would the power efficiency scale linearly? Should the FSR be modified? If the number of comb lines is increased, the bandwidth of the photonics devices (starting with grating couplers) should be increased. How scalable is this bandwidth increase? Also, for a larger kernel, does the optical loss of the MRR limit the scalability?

b. To utilize the true speed of optical computing, in practice, deep networks are needed. How can the implemented system be scaled to a deep network? How would the non-linearity be incorporated?

c. In general, how would this architecture be scaled to a large number of neurons and layers? How does the power consumption and area scale?

5. Noise – I suggest providing a more detailed noise analysis for the detection system as the SNR could affect the classification accuracy.

6. Table of comparison – the table entries should be corrected to provide a better comparison with other works:

a. Source: it is not clear what the source is referring to. The source in this work is implemented but it seems its power consumption and area are not correctly accounted for. Some other works in the table are accounting for the power consumption of the source.

b. The table implies that different works follow a similar architecture. For example, data loading (modulator array) is not required for some of the works in the table but it seems it is considered as a negative aspect for those works (same goes for modulation rate).

c. The Efficiency numbers should be corrected based on comment 3 above. Also, some other works in the table include non-linearity and/or contain multiple layers. This should be noted.

d. I suggest modifying the computing density for this work per comment 2 above. Also, there is a note that only the area of the computing core is considered. In this case, the computing core in this work should include the digital processing unit as well. Some other works in the table perform part of the computation (that this work does on the digital processor) within their computing core area (such as nonlinear processing). I suggest using only the linear compute area for all works for a fair comparison and add a note accordingly.

7. Other questions and comments:

a. Please comment on the effect of process variations on the delay lines. Would variations in different delay lines result in classification inaccuracies?

b. Please comment on the thermal cross-talk between the heaters.

c. What is the architecture of the fully connected layers implemented digitally? What hardware is used for the digital implementation?

d. How long does it take to input the image to the system?

e. How does the accuracy, classification time, and power consumption compare with a typical all-electrical hardware used for hand-written digit classification?

Reviewer #2 (Remarks to the Author):

I already reviewed this work for a different journal and I see that the authors addressed my main concerns in this resubmission. I have a few more comments for further improvements.

1. The claimed computing density is still misleading, in my perspective. The PPU, performing convolution operations, is composed of the comb source, the weight bank chip, and subsequent photo-detectors—the computing operations cannot be achieved without either of them. As such, the calculation of computing density should take all of those components' footprints into consideration.

2. More experimental details are necessary to be presented. (a) The optical and electrical spectrum of the chip's output signal (and the input signal's electrical spectrum) should be presented, to verify the operation of the MRR weight bank with data loaded. (b) A more complicated image (widely used in image processing community) should be used to demonstrate convolution operations with the Roberts operator, to make the results more impressive. (c) More experimental waveforms (and details in contrast to calculated results) should be presented, such as the results for convolution with W , W' and W'' .

3. The fully connected layer seems quite large—large enough to perform the MNIST task without the need of the convolutional layer. I suggest the authors to have a test that directly use the fully connected layer to perform the MNIST task, if it still works, then the neural network's architecture needs to be tailored to make the convolution operations useful.

4. Impacts of the MRR's relatively narrow bandwidths should be discussed. The authors employed the classic MRR array as a wavelength demultiplexer. MRRs has narrow passbands with relatively sharp roll-off rates, which work well for narrow-band operations, such as selecting single wavelengths. In this work, the MRRs are used to filter out wavelength channels with loaded data at ~ 17 GBaud, corresponding to a total bandwidth of >20 GHz with double-sideband modulation. As such, while the channel locates at a specific point of the sharp slope of the MRR, the signal suffers from huge distortions and losses, which could severely limit practical applications.

5. The employed chip, I suppose, is the same as the one used in [31]. I suggest to acknowledge this in the manuscript.

Response letter

We are grateful to the reviewers for their time and expertise in reviewing our manuscript. We have modified the manuscript in accordance with their comments and suggestions. Here, we present a point-by-point reply (in blue) to the reviewers' comments, as well as the corresponding modifications in our main manuscript and supplementary information (in red).

Response to Reviewer #1

Comment 1:

Novelty - While this work shows an interesting engineering demonstration, the novelty should be explained more clearly. Multi-wavelength photonic computing systems have been demonstrated before. Integrated Kerr combs as multi-wavelength sources (e.g. for multi-wavelength communication) have been demonstrated before. In the abstract, the claim that "both the multi-wavelength source and the fast processing core are integrated" is not accurate. They are separately implemented and coupled through an isolator and an EDFA. Please elaborate on the novelty. I suggest adding a sentence on convolution implementation method (as a novelty aspect) to the abstract.

Reply:

Thanks for this valuable suggestion.

We agree with the reviewer. The key novelty of this work is the realization of time-wavelength plane stretching convolution by a microcomb-driven chip-based photonic processing unit. While not all the components are integrated on the same substrate, standard packaging can easily assemble the whole system together in a compact version, which is compatible with current photonic industry. This shows the possibility of deploying this technology for practical applications in a massive way in the future. We agree with the reviewer that the novelty should be explained more clearly and accurately. In the revised manuscript, the novelty description in the abstract is modified as "In this work, convolution based on time-wavelength plane stretching approach is implemented on a microcomb-driven chip-based photonic processing unit (PPU), which is compatible with production in photonic industry."

Comment 2:

Performance - The characterization of performance is somewhat inaccurate. The authors indicate 1 Tops/mm² is achieved. However, a true metric is classification time. In the implemented system, a single linear photonic computation layer is implemented, which could improve the overall compute time, but the bottleneck could be the digital processing (backend). What is the true overall compute/classification time? Does the reported 5.83×10⁶/s include the digital backend? What is the equivalent Tops/mm² for overall compute/classification? Is the chip area of electronic control blocks, the microcomb and the pump laser included in the compute density calculations? In the abstract, the claim that the system can be used for "video action recognition" should be quantitatively explained. For video classification a deep network is often needed. Given that the implementation is a single linear photonic layer, and digital backend is needed, is this system suitable for video classification? Please provide quantitative justification.

Reply:

We agree with the reviewer that the bottleneck to further improve the overall compute time is the

digital processing. The structure of the CNN we used for handwritten digits classification consists of one convolutional layer with 3 kernels and two fully-connected layers. The sampling rate is 17/4 GHz and the original kernel matrix W is decomposed into two matrices W' and W'' . Since W'' is a matrix with all elements equal to 1 for all inputs, the duration to yield 3 feature maps is $27 \times 27 \times 4 / (17 \text{GHz}) \times 4 = 686.1$ ns. The fully-connected layers with two massive weight matrices (512×2187 and 10×512) are implemented on a computer. The run time of the digital backend (directly given by our computer: Intel(R) Core(TM) i7-10700K CPU) for one picture is about 5390.6 ns. Therefore, the overall compute/classification time is 6076.7 ns. The reported computing speed $5.83 \times 10^6/\text{s}$ does not include the digital backend and it should be revised as $1.65 \times 10^5/\text{s}$.

Compute density is a figure-of-merit to benchmark the power-performant multiply-accumulate operations in photonic neural networks, which is defined in ref[*IEEE Journal of Selected Topics in Quantum Electronics* 26, 1–18 (2019)] previously:

$$D = \frac{\text{Speed(MACs/s)}}{\text{Area per MAC unit(mm}^2\text{)}}$$

The unit of the speed can be multiply-accumulate operations/s (MACs/s) or operations/s (OPS) and the area refers to the unit area that perform MACs or operations. According to this definition, only the chip area where MACs are performed is included in some articles [*Nature* 589, 52–58 (2021), *Nature* 606, 501–506 (2022)]. The light sources (including pump laser or optical frequency combs), data loading section, photo-detector, electronic control blocks, etc. are excluded, although these devices are indispensable components to implement convolution calculations. Therefore, the evaluation of compute density in our previous manuscript follows the same protocol.

We agree with the reviewer that the light sources, data loading section, photo-detectors, etc. are essential components to perform convolution. Hence, the concept of compute density in our case should be explained as linear compute density of photonic core or photonic core compute density for short. In our revised manuscript and supplementary information, both the photonic core compute density and the overall compute density (the area of DFB pump laser, comb source, silicon weight bank chip and photo-detector are considered) are provided to make a relatively fair and comprehensive comparison. As a prototype, the electrical parts (including arbitrary signal generator, matrix tuning controller, oscilloscope and digital backend) and some photonic devices (such as EDFA and band-pass filter) of our proposed PPU are discrete components, which can hardly give the exact area. Therefore, in the overall compute density estimation, these areas are not included.

As the reviewers suggested, the areas of DFB pump laser, comb source, silicon weight bank chip and photo-detector, are considered to give a relatively reasonable overall compute density. Note that the photo-detector we used is a discrete commercial one. The area of the photo-detector chip in the package is difficult to provide. Therefore, the area of photo-detector is given by a vertical epitaxial Ge-Si PD which is used in our previous work [*Nature* 605, 457–463 (2022)].

Table RI. Estimated overall compute density of the prototypical PPU

Components	Footprint (mm ²)	Overall compute density (TOPS/mm ²)
DFB laser	$1.2 \times 0.33 = 0.396$	$2 \times 4 \times 17 \times 10^9 / 1.311 \approx 0.104$
Microcomb	$0.288 \times 0.288 \approx 0.083$	

Weight bank chip	$2.6 \times 0.24 + 0.82 \times 0.16 \approx 0.755^a$
Photo-detector ^b	$0.28 \times 0.275 = 0.077$
Total chip area	1.311

^a footprint of the silicon EOM and four MRRs with delay lines.

^b replaced by a common on-chip Ge-Si photo-detector

The overall compute density of the proposed PPU is estimated in the table RI. The footprints of DFB pump laser, comb source, silicon weight bank chip and photo-detector are all taken into consideration. The total chip area is about 1.311 mm² and the corresponding overall compute density is about 0.104 TOPS/mm².

Multiply-accumulate (MAC) is the fundamental operation in matrix multiplication and convolution. In principle, an architecture that can perform convolution can also conduct matrix multiplication. For example, a matrix multiplication with a weight matrix size of 64×64 can be decomposed into $64 \times 64 / 4 = 1024$ times of convolutions when the kernel size is 2×2. It is true that a deep convolutional network is often needed for complicated tasks (such as video action recognition). For example, in a convolutional neural network for human action [Journal of Manufacturing Systems 56, 605-614 (2020)], the structure of the spatial and temporal streams have 2 and 4 convolutional layers, respectively. The two-stream CNN has 128 kernels with size of 3×3, 80 kernels with size of 5×5 and 64 kernels with size of 9×9. By increasing the kernel number, kernel size and reusing the proposed PPU, the linear operations (convolution and matrix multiplication) in such algorithm can be accelerated as well.

Changes made in this revision:

(Manuscript page 8, line 29-44) “The conformable results indicate that the cross talk and noise has limited impact on the PPU performance after calibration. The structure of the CNN we used consists of one convolutional layer with 3 kernels and two fully-connected layers. The sampling rate is 17/4 GHz and the original kernel matrix W is decomposed into two matrices W' and W'' . Since W'' is a matrix with all elements equal to 1 for all inputs, the duration to yield 3 feature maps is $27 \times 27 \times 4 / (17 \text{GHz}) \times 4 = 686.1$ ns. The run time of the digital backend (directly given by our computer: Intel(R) Core(TM) i7-10700K CPU) to implement the fully-connected layers is about 5390.6 ns. Therefore, the overall classification time is 6076.7 ns. Therefore, the PPU can process handwritten digit images at the speed of $1/6076.7$ ns = 1.65×10^5 /s.”

Comment 3:

Power consumption – The power consumption calculations should be modified:

- First, the power consumption should be calculated considering the wall-plug efficiency (WPE) of the pump laser and the power consumption of the pump temperature controller. From Supp. material, it seems that the on-chip optical power of the pump is used for power consumption calculations which is not accurate. I suggest re-calculating the power consumption considering the WPE of the pump laser and the power consumption of the temperature control system.*
- The power consumption of the EDFA (shown in Fig. 3) should be considered as it is part of the system.*
- The power consumption of modulator drivers used for MZM drive should be considered.*

d. It seems a fast sampling scope is used as a part of the detection system. What is the plan to replace the scope for a practical use? What would be the power consumption overhead? Similarly, programmable power supplies are used.

e. It is inaccurate to consider the power consumption of different electronic blocks based on best reported numbers in the literature (for TIA, ADC, DAC, ...) for the reported system. First, these blocks are implemented on different processes (14nm CMOS FinFET, 7nm CMOS, ...) and there is no guarantee that they will yield the same results if implemented on the same process. Second, once connected together (either monolithically or through hybrid packaging), the loading effect, the noise effect, the coupling and interference effects most likely require re-design of these blocks and most likely results in substantial increase in the power consumption. Third, these blocks are not part of the measurement systems. The power consumption of the actual components used in the measurement setup should be considered. Also, what is the power consumption of the digital backend? It should be included in the calculations as well.

Reply:

According to our investigation, the power consumption estimations in comb based optical computing systems are ambiguous in some high-level articles. For example, in ref[*Nature 589, 44–51 (2021)*], the power consumption is estimated according to a fully integrated optical CNN with the same network configuration, although the actual optical CNN was demonstrated with discrete devices. Besides, in ref[*Nature 589, 52–58 (2021)*], only the optical power (not the power consumption of the pump laser), the analogue-to-digital converters and modulators are taken into consideration. The power consumption of EDFA, modulator drivers, TEC and the digital backend, etc. are usually excluded in the power consumption estimation.

Following this protocol, the concept of power consumption in our previous manuscript should be explained as expected power consumption. Benefit from the recent advances in integrated optical frequency comb[*Nature 562, 401–405 (2018)*, *Nature Photonics 16.2, 95-108 (2022)*], circuit-based erbium-doped amplifier[*Science 376, 1309–1313 (2022)*] and the mature on-chip filter technology, the discrete EDFA, band-pass filter can be replaced by fully integrated photonic devices eventually. Furthermore, with the development of hybrid and monolithic integration technology, the light source, silicon photonic circuit and the associated electronic blocks (driver, TIA, ADC, DAC, etc.) can be all integrated in a same mainboard or even in a single chip. The expected power consumption shows the ultimate potential of our architecture, although there is still a long way to go.

Table RII. Estimated power consumption of the proof-of-concept PPU

Components	Voltage (V)	Current (A)	Power (W)
DFB laser	1.500	0.300	0.450
Silicon photonic chip			
DC bias1	2.332	0.047	0.109
DC bias2	1.697	0.034	0.058
Thermal phase shifter1	0	0	0
Thermal phase shifter2	2.500	1.25×10^{-2}	0.031
On ring heaters	~2	$\sim 5 \times 10^{-3}$	$\sim 0.01 \times 4 = 0.04$
EDFA	N/A	0.860	$\sim 50^a$

Modulator driver1	9.000	0.275	2.475
Modulator driver2	9.000	0.258	2.322
TEC for DFB laser	1.830	0.680	1.244
TEC for microcomb	1.372	0.078	0.107
TEC for Silicon photonic chip	5.920	0.450	2.664
CPU			~30 ^b
Total power consumption			~89.5

^a typical power consumption according to the user manual.

^b from a power monitor software[<https://www.hwinfo.com/>] when implementing fully-connected layers.

We agree with the reviewer that the actual components used in the measurement setup should be considered. Therefore, the current power consumption (including the consumption of the EDFA, TEC and the digital backend, etc.) of our proof-of-concept PPU are provided in our revised manuscript and supplementary information. However, to make a relatively fair comparison with other works (especially the optical computing architectures based on microcombs), both the expected efficiency calculated with the similar method in ref[*Nature 589, 44–51 (2021)*, *Nature 589, 52–58 (2021)*] and the current power consumption calculated according to the actual components used in our measurement setup are provided in the supplementary information. The detailed calculation process is shown in supplementary information IX, as follows:

“The power consumption of the current PPU are mainly from six aspects: InP DFB pump laser, EDFA, silicon photonic chip, modulator drivers, TEC and digital backend, as listed in table SIII.

Table RIII. Estimated power consumption of the proof-of-concept PPU

Components	Voltage (V)	Current (A)	Power (W)
DFB laser	1.500	0.300	0.450
Silicon photonic chip			
DC bias1	2.332	0.047	0.109
DC bias2	1.697	0.034	0.058
Thermal phase shifter1	0	0	0
Thermal phase shifter2	2.500	1.25×10^{-2}	0.031
On ring heaters	~2	$\sim 5 \times 10^{-3}$	$\sim 0.01 \times 4 = 0.04$
EDFA	N/A	0.860	~50 ^a
Modulator driver1	9.000	0.275	2.475
Modulator driver2	9.000	0.258	2.322
TEC for DFB laser	1.830	0.680	1.244
TEC for microcomb	1.372	0.078	0.107
TEC for Silicon photonic chip	5.920	0.450	2.664
CPU			~30 ^b
Total power consumption			~89.5

^a typical power consumption according to the user manual.

^b from a power monitor software[19] when implementing fully-connected layers.

For the Silicon photonic chip, the DC bias represents the reverse bias voltage for depleted PN junction of the Si modulator; The thermal phase shifters are used to adjust the demanded operating status for DC bias-point of the EO modulator; The kernel weights are tuned by the 4 on ring heaters. The power consumption information of the CPU to implement fully-connected layers is directly obtained from a power consumption monitor software[19]. The total current power consumption is calculated as about 89.5 W, including the power consumption of the EDFA and digital backend.

Benefit from the recent advances in integrated optical frequency combs[4, 20], circuit-based erbium-doped amplifier[21], on-chip semiconductor optical amplifiers (SOA)[22] and the mature on-chip filter technology, the discrete EDFA, band-pass filter can be replaced by fully integrated photonic devices eventually. Furthermore, with the development of hybrid and monolithic integration technology, the light source, silicon photonic circuit and the associated electronic blocks(including modulator drivers, transimpedance amplifiers (TIAs), digital-to-analog converters (DACs), analog-to-digital converters (ADCs)) can be all integrated in a same main-board or even in a single chip.

To show the ultimate potential of our architecture, the expected power consumption calculated in line with similar protocols in[8, 23] is also provided here. The on-chip pump power to generate the microcomb can be as low as 98 mW[4]. When using low-loss phase change materials[24], the power consumption of the MRR weight bank could be "near zero". The typical power consumption of the on-chip SOA[22] is 390 mW. The energy consumption from photodetection is dominated by the transimpedance amplifier (TIA). The power cost from the assorted digital circuits: 5.36 pJ/Sa driver (28G Hz)[25], 1.14 pJ/Sa TIA (53G Hz)[26], 2.72 pJ/conversion DAC (8 bit)[27], 2 pJ/conversion ADC (8 bit)[28]. If utilizing high order MRRs[29] combined with MZI-embedded microring[30], the flat-top pass band enable high modulation rate and the weight tuning can be realized by the embedded MZI. The computing speed of the prototypical PPU is 0.136 TOPS and the expected power consumption is $98+390+(5.36+1.14+2.72+2)\times 17=678.74$ mW. The expected energy efficiency of our prototypical PPU is $0.136 \text{ TOPS}/678.74 \text{ mW} \approx 0.2 \text{ TOPS/W}$.

For a PPU with 5×5 kernel matrix size, if the modulation rate is 50 Gbaud, the computing speed of the PPU will be promoted to $50\times 10^9\times 25\times 2=2.5 \text{ TOPS}$. The corresponding expected power consumption will be $98+390+(5.36+1.14+2.72+2)\times 50=1049 \text{ mW}=1.049 \text{ W}$. Then, the expected energy efficiency will be $2.5 \text{ TOPS}/1.049 \text{ W} \approx 2.38 \text{ TOPS/W}$. It should be noted that the expected power consumption estimation is based on ideal conditions. The performance of different electronic blocks may not the same if implemented on a single processor and many factors (such as thermal, coupling, interference, etc.) can lead to an increase in the power consumption once they are packaged together. Nevertheless, compared to its electronic competitors (e.g. CPU and GPU), the potential energy consumption advantage of the PPU is still significant."

The actual power consumption of our proof-of-concept computing system derives from the InP DFB laser, EDFA, silicon photonic chip, modulator drivers, TEC and digital backend, as listed in Table RII.

For the Silicon photonic chip, the DC bias represents the reverse bias voltage for depleted PN junction of the Si modulator; The thermal phase shifters are used to adjust the demanded operating status for DC bias-point of the EO modulator; The kernel weights are tuned by the 4 on-ring heaters. The power consumption information of the CPU to implement fully-connected layers is directly obtained from a power consumption monitor software [<https://www.hwinfo.com/>].

- a. We agree that power consumption should be calculated considering the wall-plug efficiency (WPE) of the pump laser and the power consumption of the pump temperature controller. In our experiment, the DFB pump laser is packaged with a TEC module (Phononic FBM-013865). The re-calculated power consumption of the pump laser with TEC is $0.45+1.244\approx 1.69$ W.
- b. As a part of the computing system, the power consumption of the EDFA should be considered. Because we can only get the current information from the panel of the instrument, the power consumption of the EDFA cannot be calculated accurately. Therefore, we estimate the power consumption as about 50 W according to the user manual.
- c. The power consumption of the two modulator drivers used for MZM drive is $2.475+2.322\approx 4.80$ W
- d. We did use a fast real-time oscilloscope as a part of the detection system for a proof-of-concept. The power consumption of the real-time oscilloscope is between 1300 W to 2600 W, according to the instrument manual. It is too high for practical applications. One possible solution to replace the scope is utilizing a fast analog-to-digital converter chip with custom-developed high-speed circuits in the future. As the power supplies are used for DC control and powering the modulator drivers, the power consumption of the DC part can be estimated as $0.109+0.058+0.031+0.04+2.475+2.322\approx 5.04$ W.
- e. We agree with the reviewer that the power consumption of the actual components used in the measurement setup should be considered. Table SII shows the calculation details according to the actual components we used. The power consumption is calculated as about 89.5 W, including the power consumption of the EDFA and digital backend. In our revised manuscript, the concept of power consumption is divided into expected energy consumption and the energy consumption of the current proof-of-concept PPU. The significance of discussing expected energy consumption in line with the similar estimation protocol in ref.[*Nature* 589, 44–51 (2021)] is to reflect the ultimate potentials of our architecture, although there is still a long way to go. Nevertheless, compared to its electronic competitors (e.g. CPU and GPU), the potential energy consumption advantage of the PPU is still significant.

Comment 4:

Scalability – There are multiple aspects to the scalability of this work:

- a. *Microcomb: the design is very nice. For scalability to a larger number of lines ($n>4$), corresponding to a larger kernel size, it is not clear that a larger number of lines at the same per-line power could be generated. What is the effect of unequal power of comb lines on the classification accuracy and system power consumption? Would the power efficiency scale linearly? Should the FSR be modified? If the number of comb lines is increased, the bandwidth of the photonics devices (starting with grating couplers) should be increased. How scalable is this bandwidth increase? Also, for a larger kernel, does the optical loss of the MRR limit the scalability?*
- b. *To utilize the true speed of optical computing, in practice, deep networks are needed. How can the implemented system be scaled to a deep network? How would the non-linearity be incorporated?*
- c. *In general, how would this architecture be scaled to a large number of neurons and layers? How does the power consumption and area scale?*

Reply:

- a. Due to the non-flat optical spectrum of the microcomb we used, the same per-line power cannot

be generated when the number of comb lines is too large. Inputting the unequal comb lines directly to the MRR weight bank is equivalent to preloading a random weight matrix before the convolution calculation, which will deteriorate the classification accuracy. In our experiment, we chose a relatively flat region of the comb spectrum and the power of the comb lines is calibrated to the same by minor-tuning the resonant peak positions of the microrings. For a larger kernel size, the low-power comb lines will reduce the signal-to-noise ratio of the received signal, which also deteriorate the classification accuracy.

In our proposed PPU, considering the number of comb lines that can be used (low power comb lines are excluded) and the limited input power to the grating coupler, the power efficiency scales linearly only when the size of the kernel is relatively small. Therefore, the kernel size is limited to 5×5 in the discussion about energy efficiency in our manuscript. However, large kernel size is attainable if using larger AlGaAs microrings or high-Q silicon nitride (Si_3N_4) resonators [*Nature* **546**, 274-279 (2017)] with wide flat comb spectrum region (C+L band) and reducing the insertion loss of the optical computing system.

The FSR of the microring determines the number of comb lines that can be used, that is, the kernel size. By decreasing the radius of the microring, using two cascaded microrings with vernier effect [*Optics Communications* **284**, 156-159 (2011)], or designing FSR-free microring with contra-directional couplers [*Optics express* **24**, 29009-29021 (2016)], the FSR can be expanded for a larger kernel size. We agree that the bandwidth of the photonics devices should be increased if using more comb lines. In theory, the bandwidth of the photonics devices scale linearly as the number of comb lines increase. For example, if the number of comb lines is 5×5 and the wavelength interval is 1.5 nm, the optical bandwidth of the grating couplers, EO modulator and photo-detector should be at least 37.5 nm. However, by redesigning the microcomb, the wavelength interval can be reduced appropriately according to the actual modulation rate. The grating couplers can be replaced by edge couplers with high optical bandwidth. Through optoelectronic co-design, doping concentration/distribution optimization and electrode structure optimization, the optical bandwidth of EO modulator and photo-detector can be further broadened. Fortunately, however, the 3×3 or 5×5 kernel size can already meet the computing requirement of most convolutional neural networks. Kernels with large size are usually not necessary.

Not only the optical loss of the MRR, but also the loss of the grating couplers and EO modulator limit the scalability. Due to the nonlinear effects of the silicon waveguide, such as two-photon absorption and free carrier absorption, the total optical input power is limited. In order to ensure that the waveform is not distorted and the photo-detector can respond to the optical signal, the insertion loss should be minimized for a large-scale photonic computing system.

b. It is true that a deep convolutional network is often needed for complicated tasks. It is difficult for our system to directly map a deep neural network with so many layers because of the computing mechanism. Even so, by increasing the kernel number, decomposing large matrix or kernel into small ones and reusing the proposed PPU, the linear operations in deep networks can be accelerated as well.

Nonlinear activation is an indispensable component in neural networks. Implementing nonlinear activation in optical domain for our system is quite challenging because of the information loading method and the temporal relationship of different wavelength signals. One possible solution is utilizing a limited trans-impedance amplifier (TIA) to simulate ReLU nonlinear activation function. We

will explore the feasibility of this approach in future work.

c. As shown in Fig.1 in the main manuscript, the architecture can be scaled to a large number of neurons by utilizing more comb lines and increasing the kernel size. As mentioned above, implementing nonlinear activation unit in optical domain for our system is quite challenging, further exploration is needed to realize a multi-layer neural network. For the current hardware architecture, the power consumption mainly comes from the modulator drivers, EDFA, TECs and digital backend. Given that the of the EDFA, TECs and digital backend does not change much, the power consumption increment will show trend with the increase of the number of modulators (that is, the number of convolution kernels). Since the footprint of the EO modulator accounts for a large proportion of the total chip area and the number of pump laser and microcomb does not change, the area of the architecture also scales rough linearly as the number of modulators increases.

Comment 5:

Noise – I suggest providing a more detailed noise analysis for the detection system as the SNR could affect the classification accuracy.

Reply:

To emphasize the influence of the SNR on the classification accuracy, the hidden layer (with 512 neurons) in the fully-connected layers is removed. The noise in the PPU mainly includes phase noise and ASE noise, which can be described by gaussian distribution. Here, we add white gaussian noise to the ideal feature maps to simulate the detected signals. The noise power level is calculated according to the SNR, as shown in the equation below,

$$P_{noise} = P_{signal} \times 10^{SNR(dB)/10}$$

where P_{noise} and P_{signal} are the power of noise and signal, respectively.

Fig. R1. Classification accuracy vs. SNR

As shown in Fig. R1, the classification accuracy rises as SNR increases. At very low SNR, the noise has significant impact on the waveforms of the feature maps and the neural network can hardly classify the digits to correct categories. At SNR = 10 dB and beyond, the classification accuracy oscillates and finally converges to 91.2%. Some classification accuracy is higher than the theoretical result (91.2%).

One possible reason is that the neural network trained from noise-free samples has good generalization ability to correctly classify digits with moderate noise. At very high SNR, such as 35 dB, the noise power is so small that the feature maps can be regarded as noise-free.

Furthermore, the SNR calculated from the real experimental data is 17.65 dB. Without the hidden layer, the classification accuracy of the PPU with digital backend is 88.2%, which is a little inconsistent with our simulation. The accuracy degradation comes from multiple factors, not only the SNR. The first factor is the waveform distortion caused by the relatively sharp roll-off rates of the MRR. As the frequency response of MRR is not flat, it distorts the output signals as the high frequency components may be attenuated. The second reason is the inherent non-linear transmission function of the silicon MZI modulator, which cannot provide the perfect transformation from the input RF signals to optical signals. Therefore, it is reasonable that the accuracy from the real data is lower than the simulation.

Comment 6:

Table of comparison – the table entries should be corrected to provide a better comparison with other works:

a. Source: it is not clear what the source is referring to. The source in this work is implemented but it seems its power consumption and area are not correctly accounted for. Some other works in the table are accounting for the power consumption of the source.

b. The table implies that different works follow a similar architecture. For example, data loading (modulator array) is not required for some of the works in the table but it seems it is considered as a negative aspect for those works (same goes for modulation rate).

c. The Efficiency numbers should be corrected based on comment 3 above. Also, some other works in the table include non-linearity and/or contain multiple layers. This should be noted.

d. I suggest modifying the compute density for this work per comment 2 above. Also, there is a note that only the area of the computing core is considered. In this case, the computing core in this work should include the digital processing unit as well. Some other works in the table perform part of the computation (that this work does on the digital processor) within their computing core area (such as nonlinear processing). I suggest using only the linear compute area for all works for a fair comparison and add a note accordingly.

Reply:

a. The source in the table refers to the optical source. In our revised manuscript and supplementary information, the power consumption of the optical source is accounted for.

b. To make a comprehensive comparison, a variety of architectures are listed in the table. Due to the differences in optical computing mechanics, it is indeed inappropriate to include all of them in a unified framework for comparison. In the spatial diffraction approach, for example, the modulator array is not always required and it is inappropriate to use compute density (a figure-of-merit for integrated photonic computing hardware) for performance evaluation. In our revised supplementary information, the comparison is focused on the integrated photonic computing architectures that can perform convolution or matrix multiplication and modulator (for data loading) is an optional part.

c. We agree with the reviewer that the actual components used in the measurement setup should be considered. Therefore, the current power consumption (including the consumption of the EDFA, TEC

and the digital backend, etc.) are provided in our revised manuscript and supplementary information. However, to make a relatively fair comparison with other works (especially the optical computing architectures based on microcombs), both the expected efficiency calculated with the similar method in ref[*Nature 589, 44–51 (2021), Nature 589, 52–58 (2021)*] and provided in the supplementary information. The detailed calculation process is shown in supplementary information IX. The works include non-linearity and/or contain multiple layers are noted in the comparison table.

d. Thanks for the suggestion. We agree with the reviewer that the area of the computing core should include the digital processing unit. According to the official information provided by Intel[<https://www.intel.com/content/www/us/en/products/sku/199335/intel-core-i710700k-processor-16m-cache-up-to-5-10-ghz/specifications.html>], the package size of the chip is 37.5mm x 37.5mm. However, the area of the chip itself and the area of each functional region (especially the computing core) have no public information that can be queried. Therefore, the chip area of the digital processing unit is not included.

As replied in comment 2 above, the conventional method is to consider only the chip area that conduct photonic linear operations when evaluating the compute density of a photonic computing system. In our revised main manuscript and supplementary information, both the photonic core compute density and the overall compute density which considering the footprints of DFB pump laser, comb source, silicon weight bank chip and photo-detector are provided. We also added a note that “only the photonic linear operation area is considered for all works” for a fair comparison in Supplementary note VIII.

Changes made in this revision:

(Supplementary note VIII: Compute density evaluation)

To compare the performance of various photonic computing architectures, we define a figure-of-merit: photonic core compute density as follows[7]:

$$\text{Photonic core compute density} = \frac{\text{Computing speed (TOPS)}}{\text{Area of photonic linear operations unit (mm}^2\text{)}}$$

In this metric (also in line with the estimation protocols in ref[8, 9]), the photonic core compute density of our proof-of-concept PPU is 1.04 TOPS/mm². As the pump laser, microcomb source, silicon weight bank chip, and subsequent photo-detector are all essential components to perform convolution operations, here, we define another figure-of-merit: overall compute density as follows:

$$\text{Overall compute density} = \frac{\text{Computing speed (TOPS)}}{\text{Total area of the photonic chips (mm}^2\text{)}}$$

where the footprints of the mentioned above are all taken into consideration. To make a relatively comprehensive comparison, the detailed overall compute density estimation is shown in Table S I.

Table S I: Overall compute density estimation of the prototypical PPU

Components	Footprint (mm ²)	Overall compute density (TOPS/mm ²)
DFB laser	1.2×0.33=0.396	2×4×17×10 ⁹ /1.311 ≈ 0.104

Microcomb	$0.288 \times 0.288 \approx 0.083$
Weight bank chip	$2.6 \times 0.24 + 0.82 \times 0.16 \approx 0.755^a$
Photo-detector ^b	$0.28 \times 0.275 = 0.077$
Total chip area	1.311

^a footprint of the silicon EOM and four MRRs with delay lines.

^b replaced by a common on-chip Ge-Si photo-detector

The footprints of DFB pump laser, comb source, silicon weight bank chip and photo-detector are all taken into consideration. The total photonic chip area is about 1.311 mm² and the corresponding overall compute density is about 0.104 TOPS/mm². Strictly speaking, the area of the digital processing unit (Intel(R) Core(TM) i7-10700K CPU) should be included in the compute density estimation. However, the area of the digital chip and the area of each functional region (especially the computing core) have no public information that can be queried. Therefore, the area of the digital processing unit is not included in this estimation.

Table S II: Comparison of state-of-the-art integrated photonic computing hardwares

Technology	Data loading rate	Weight precision	Energy efficiency	Compute density ^a
	(Gband)	(bits)	(TOPS/W)	(TOPS/mm ²)
MZI mesh[10]	—	8	—	—
MZI mesh[11]	1×10^{-5}	4	—	9.88×10^{-7}
Cascaded MZI[12]	1×10^{-7}	6.6	—	—
InP SOA[13]	10	~4.5	~0.24	—
Diffractive cell[14]	$\sim 1 \times 10^{-5}$	—	~0.11	$\sim 3.77 \times 10^{-3}$
MRRs[15] ^b	0.047	>5.5	—	—
Photonic neuron[9] ^b	N/A	—	~0.07	3.5
PCM+Waveguide[16]	1×10^{-6}	6	—	2.5×10^{-6}
WDM+PCM[8]	2	5	0.4	0.2
This work	17	9	$0.2/1.52 \times 10^{-3}$ ^c	$1.04/0.104^d$

^a only the photonic linear operation area is considered for all works.

^b contains multiple layers.

^c refers to expected/current energy efficiency.

^d refers to photonic-core/overall compute density.

A comparison among the representative integrated photonic computing hardware is summarized in Table II. Due to the high data loading rate and the associated calibration procedure, the photonic core compute density in this work is over 1TOPS/mm² and a record high weight control precision of 9 bits is achieved. Although the overall compute density of the prototypical PPU is small, it can be further improved by increasing the modulation speed and reducing the chip area. For instance, using a monolithic integrated microcomb source[17] and replacing the silicon modulator with ultra-compact hybrid plasmonic Mach-Zehnder modulator[18], the total chip area can be dramatically reduced.

Comment 7:

Other questions and comments:

a. Please comment on the effect of process variations on the delay lines. Would variations in different delay lines result in classification inaccuracies?

b. Please comment on the thermal cross-talk between the heaters.

c. What is the architecture of the fully connected layers implemented digitally? What hardware is used for the digital implementation?

d. How long does it take to input the image to the system?

e. How does the accuracy, classification time, and power consumption compare with a typical all-electrical hardware used for hand-written digit classification?

Reply:

a. For a MRR weight bank with $n-1$ delay lines, assuming the modulation rate is f_x and the variations on the i th is Δt_i , the condition that intersymbol crosstalk does not occur is

$$\begin{cases} (n-1)T + \sum_{i=0}^{i=n-1} \Delta t_i < (n-1)T + \frac{1}{2}T \\ nT + \sum_{i=0}^{i=n-1} \Delta t_i > (n-1)T + \frac{1}{2}T \end{cases} \rightarrow \left| \sum_{i=0}^{i=n-1} \Delta t_i \right| < \frac{1}{2}T \quad (S1)$$

where $T=1/f_x$. Theoretically speaking, if the total time delay variations is smaller than $0.5T$, the variations in different delay lines has little impact on the classification accuracy.

b. The thermal crosstalk happens when the local heat leaks to the surroundings during operation, leading to resonance shifts in the nearby MRRs. Although thermo-electric cooler (TEC) are valid approaches to reduce the thermal crosstalk, the thermal crosstalk still exist when tuning other channels. to solve the adverse effects of inner cross-talk (including thermal crosstalk), a calibration procedure for MRR weight bank using in-situ GDC method is developed and weight control accuracy as high as 9 bits is achieved.

c. To prove the validity of the convolutional layer, the fully-connected layer implemented digitally in our experiment consists of two weight matrices with the size of 512×2187 and 10×512 . The architecture of the fully connected layer is shown in Fig. 4a in the main manuscript. The hardware used for the digital implementation is Intel(R) Core(TM) i7-10700K CPU.

d. For handwritten digits classification, the size of the input image is 28×28 . Since the kernel size is 2×2 and the stride is 1, the size of the total input vector is $(27 \times 27 \times 2 \times 2 = 2916, 1)$. For one image, the time it takes to input to the system is $2916 / (17\text{GHz}) \approx 171.53 \text{ ns}$.

e. The comparison between our proposed PPU and an all-electrical hardware (Intel(R) Core(TM) i7-10700K CPU) used for hand-written digit classification is list in the table below.

Table RIII. Comparison between our proposed PPU and an all-electrical hardware

	Accuracy (%)	Classification time (ns)	Expected/Current power consumption (W)
PPU	96.6	6076.7	1.049/89.5

As calculated previously, the duration to yield 3 feature maps of our PPU is $27 \times 27 \times 4 / (17 \text{GHz}) \times 4 \approx 686.1$ ns. The run time of the CPU to implement convolution layer and fully-connected layer (digital backend) is around 4687.5 ns and 5390.6 ns, respectively. Therefore, the total classification time of our PPU and CPU is $686.1 + 5390.6 = 6076.7$ ns and $4687.5 + 5390.6 = 10078.1$ ns, respectively. The power consumption estimation of the PPU according to the actual components is about 89.5 W. The expected power consumption evaluation in line with the similar protocols in ref. [*Nature* **589**, 44–51 (2021), *Nature* **589**, 52–58 (2021)] is about 1.05 W. It should be noted that about 90% of the energy consumption of our prototype comes from the bench-top instruments (EDFA and CPU). However, the energy efficiency of the PPU can be greatly improved with superior integration techniques and optimized photonic devices. For example, on-chip semiconductor optical amplifiers (SOA) [*IEEE Journal of Selected Topics in Quantum Electronics* **22**, 78–88 (2016)] or circuit-based erbium-doped amplifier [*Science* **376**, 1309–1313 (2022)] can replace the discrete EDFA. The digital backend can be removed by increasing the kernel number, kernel size and reusing the proposed PPU. Besides, the digital circuits, such as control unit, driver, TIA, etc. can be monolithically integrated with photonic devices [*Nature* **556**, 349–354 (2018)], which further improves the compactness and power-efficiency of the PPU. Although the temperature control and some effects in packaging (loading effect, coupling and interference effects, etc.) will cause extra power consumption, the potential energy efficiency advantage of the PPU is still significant.

Response to Reviewer #2

General comments:

I already reviewed this work for a different journal and I see that the authors addressed my main concerns in this resubmission. I have a few more comments for further improvements.

Reply:

Thanks for the constructive comments.

The insightful comments have been helpful for allowing us to clarify and further improve our manuscript. We have fully addressed the comments point-by-point as below.

Comment 1:

The claimed compute density is still misleading, in my perspective. The PPU, performing convolution operations, is composed of the comb source, the weight bank chip, and subsequent photo-detectors—the computing operations cannot be achieved without either of them. As such, the calculation of compute density should take all of those components' footprints into consideration.

Reply:

Compute density is a figure-of-merit to benchmark the power-performant multiply-accumulate operations in photonic neural networks, which is defined in ref[*IEEE Journal of Selected Topics in Quantum Electronics* 26, 1–18 (2019)] previously:

$$D = \frac{\text{Speed(MACs/s)}}{\text{Area per MAC unit(mm}^2\text{)}}$$

The unit of the speed can be multiply-accumulate operations/s (MACs/s) or operations/s (OPS) and the area refers to the unit area that perform MACs or operations. According to this definition, only the chip area where MACs are performed is included in some articles [*Nature* 589, 52–58 (2021), *Nature* 606, 501–506 (2022)]. The light sources (including pump laser or optical frequency combs), data loading section, photo-detector, electronic control blocks, etc. are excluded, although these devices are indispensable components to implement convolution calculations. Therefore, the evaluation of compute density in our previous manuscript follows the same protocol.

We agree with the reviewer that the pump laser, the microcomb, the weight bank chip, and subsequent photo-detectors are all essential components to perform convolution operations and the footprints of them should be taken into consideration for the overall compute density estimation. Hence, the concept of compute density in our case should be explained as linear compute density of photonic core or photonic core compute density for short. In our revised manuscript and supplementary information, both the photonic core compute density and the overall compute density (the area of DFB pump laser, comb source, silicon weight bank chip and photo-detector are considered) are provided to make a relatively fair and comprehensive comparison. As a prototype, the electrical parts (including arbitrary signal generator, matrix tuning controller, oscilloscope and digital backend) and some photonic devices (such as EDFA and band-pass filter) of our proposed PPU are discrete components, which can hardly give the exact area. Therefore, in the overall compute density estimation, these areas are not included.

As the reviewers suggested, the areas of DFB pump laser, comb source, silicon weight bank chip and

photo-detector, are considered to give a relatively reasonable overall compute density. Note that the photo-detector we used is a discrete commercial one. The area of the photo-detector chip in the package is difficult to provide. Therefore, the area of photo-detector is given by a vertical epitaxial Ge-Si PD which is used in our previous work [*Nature* **605**, 457–463 (2022)].

Table RI. Estimated overall compute density of the prototypical PPU

Components	Footprint (mm ²)	Overall compute density (TOPS/mm ²)
DFB laser	$1.2 \times 0.33 = 0.396$	
Microcomb	$0.288 \times 0.288 \approx 0.083$	
Weight bank chip	$2.6 \times 0.24 + 0.82 \times 0.16 \approx 0.755^a$	$2 \times 4 \times 17 \times 10^9 / 1.311 \approx 0.104$
Photo-detector ^b	$0.28 \times 0.275 = 0.077$	
Total chip area	1.311	

^a footprint of the silicon EOM and four MRRs with delay lines.

^b replaced by a common on-chip Ge-Si photo-detector

The overall compute density of the proposed PPU is estimated in the table RI. The footprints of DFB pump laser, comb source, silicon weight bank chip and photo-detector are all taken into consideration. The total chip area is about 1.311 mm² and the corresponding overall compute density is about 0.104 TOPS/mm².

Comment 2:

More experimental details are necessary to be presented.

(a) The optical and electrical spectrum of the chip's output signal (and the input signal's electrical spectrum) should be presented, to verify the operation of the MRR weight bank with data loaded.

(b) A more complicated image (widely used in image processing community) should be used to demonstrated convolution operations with the Roberts operator, to make the results more impressive.

(c) More experimental waveforms (and details in contrast to calculated results) should be presented, such as the results for convolution with W , W' and W'' .

Reply:

a. The input/out signal's electrical spectrum and the output optical spectrum of the chip are presented the revised Supplementary note VII. Fig. S6a shows the normalized optical spectrum of the chip's output when the three kennels for digits recognition (W_1' , W_2' and W_3') are implemented. Digits 0, 4, 8 are chosen as some examples. To verify the operation of the MRR weight bank with data loaded, the electrical spectrum of the chip's input/output is given in Fig. S7a and Fig. S7b.

Fig. S7: Calculated and experimental waveforms. a, Input electrical spectrum. b, The calculated and measured results for convolution with W , W' and W'' .

b. A more complicated image (house) from the USC-SIPI image database [<https://sipi.usc.edu/database/database.php?volume=misc&image=5#top>] is used to demonstrate convolution operations with the Roberts operator. The result of the image edge detection is shown in Fig.3a and Fig.S4 in the revised manuscript and supplementary information, respectively.

Fig. 3. Convolution for image processing. **a**, Schematic of photonic convolution and the experimental result of image edge detection with two 2×2 kernels (Roberts operator). A picture (house) from USC-SIPI image database is used as the input image. **b**, Working principle of the microcomb-driven photonic processing unit. The wavelength-time plane is stretched by the on-chip optical delay lines with time delay step Δt . iso, isolator; BF, band-pass filter.

Fig. S4: a, The normalized spectrum of the microcomb lines when W'_x , W'_y and W'' are implemented. b, Edge detection procedure using Roberts operator.

c. Both the experimental waveforms maps and the calculated results are presented in Fig. S7b (see Supplementary note VII). The experimental and calculated results for convolution with W , W' and W'' are well conformed.

Comment 3:

The fully connected layer seems quite large—large enough to perform the MINIST task without the need of the convolutional layer. I suggest the authors to have a test that directly use the fully connected layer to perform the MINIST task, if it still works, then the neural network’s architecture needs to be tailored to make the convolution operations useful.

Reply:

Thanks for the suggestion. Although the single convolution layer is quite small compared with the fully connected layer, it plays an important role in improving the classification accuracy and model stability. To further verify the function of convolutional layer, we perform the MINIST task with and without the convolutional layer. Table RIV shows the classification accuracy.

Table RIV. Function verification of convolutional layer

Number of times	Classification accuracy (%)			
	Without convolutional layer		With convolutional layer	
	30 epochs	50 epochs	30 epochs	50 epochs
1	94.23	74.56	87.66	97.88

2	84.90	95.50	97.17	98.1
3	94.16	76.95	97.73	98
4	85.17	94.73	97.71	98.02
5	83.25	95.50	97.61	97.87

It can be seen that the classification accuracy varies between 75% and 95% without the convolutional layer. While, it basically remains above 97% with the convolutional layer. Furthermore, to highlight the role of the convolutional layer, we remove the hidden layer (512 neurons) in the fully connected layer architecture. The confusion matrices of this digit classification text are illustrated in Fig. R1.

Fig. R2. Confusion matrices of digit classification text without the hidden layer

The experimental classification accuracy is 88.2%, which is very close to the calculated result (91.2%). The function of the convolutional layer we implemented in the optical domain is verified again.

Comment 4:

Impacts of the MRR's relatively narrow bandwidths should be discussed. The authors employed the classic MRR array as a wavelength demultiplexer. MRRs has narrow passbands with relatively sharp roll-off rates, which work well for narrow-band operations, such as selecting single wavelengths. In this work, the MRRs are used to filter out wavelength channels with loaded data at ~17GBaud, corresponding to a total bandwidth of >20GHz with double-sideband modulation. As such, while the channel locates at a specific point of the sharp slope of the MRR, the signal suffers from huge distortions and losses, which could severely limit practical applications.

Reply:

As the MRRs undertake two tasks of wavelength selecting and weight loading at the same time, we agree with the reviewer that the relatively narrow bandwidth of the MRRs in our current weight bank architecture will cause huge distortions and losses for double-sideband modulation at high data loading rate. One possible solution to this problem is utilizing high order MRRs[*Optics express 15, 11934-11941 (2007)*] combined with MZI-embedded microring[*IEEE Photonics Technology Letters 34, 436-439 (2022)*]. The flat-top pass band enable the high-speed data loading and the weight tuning are realized by the embedded MZI. For example, if the modulation rate is 50 Gbaud, the required

bandwidth is about 100 GHz, corresponding to ~ 0.8 nm wavelength interval in C band. Both the MRR weight bank with high order microrings and our 2-FSR spacing (182 GHz) microcomb can support this high speed data loading. In our future work, we will explore the approach and modify the prototypical PPU for practical applications.

Comment 5:

The employed chip, I suppose, is the same as the one used in [31]. I suggest to acknowledge this in the manuscript.

Reply: Thanks for the suggestion.

The processing core we employed in our experiments is indeed the same as the one used in Ref.[31]. We have added a description (“**the same one used in our recent work**”, page 9, line 41-42) to acknowledge this in the manuscript.

REVIEWERS' COMMENTS

Reviewer #1 (Remarks to the Author):

I would like to thank the authors for addressing my comments and questions. I have a few more comments:

- In the revised manuscript (page 9), the authors are acknowledging the fact that the photonic chip itself has been reported in reference 31 before. This should be included on pages 2 (introduction) and page 4 where the chip is discussed.
- when the compute density is discussed in the main manuscript, the compute density for both the photonic-core and the overall compute density should be included.
- For the statement added to page 9 of the revised manuscript "an expected energy efficiency of 2.38 TOPS/W for a fully integrated PPU with 50 Gbaud modulation rate and 5×5 kernel size", authors should clearly indicate what blocks are included in this efficiency calculation and what blocks are not (digital parts of network, laser, temp. controller, ...).

Reviewer #2 (Remarks to the Author):

The authors did an admirable job and fully addressed my comments. I have no further suggestions and support the publication of this work.

Response letter

We are grateful to the reviewers for reviewing our manuscript again. We have modified the manuscript in accordance with their comments and suggestions. Here, we present a point-by-point reply (in blue) to the reviewers' comments, as well as the corresponding modifications in our main manuscript and supplementary information (in red).

Response to Reviewer #1

Comment 1:

In the revised manuscript (page 9), the authors are acknowledging the fact that the photonic chip itself has been reported in reference 31 before. This should be included on pages 2 (introduction) and page 4 where the chip is discussed.

Reply:

Thanks for the suggestion.

In the modified manuscript, we have mentioned this point in the introduction part (pages 2) and where the chip is discussed (pages 4). Due to the addition of these acknowledgements, the number of the reference has changed from the original 31 to 20.

Changes made in this revision:

(Manuscript page 2, line 31-32) "...can also be used as a reconfigurable microwave photonic filter²⁰..."

(Manuscript page 4, line 55-56) "...with the same structure reported in our latest work²⁰..."

Comment 2:

When the compute density is discussed in the main manuscript, the compute density for both the photonic-core and the overall compute density should be included.

Reply:

We agree with the reviewer. In our revised manuscript, both the photonic-core and the overall compute density are included when discussing the compute density.

Changes made in this revision:

(Manuscript page 2, line 44-46) "...this PPU exhibits a preeminent photonic-core (overall) compute density of 1.04 TOPS/mm² (0.104 TOPS/mm²)..."

(Manuscript page 6, line 82-87) "...If the footprints of DFB pump laser, comb source, silicon weight bank chip and photo-detector are all taken into consideration. The total chip area is about 1.311 mm² and the corresponding overall compute density is about 0.104 TOPS/mm² (see Supplementary note VIII for details)."

Comment 3:

For the statement added to page 9 of the revised manuscript "an excepted energy efficiency of 2.38 TOPS/W for a fully integrated PPU with 50 Gbaud modulation rate and 5x5 kernel size", authors should clearly indicate what blocks are included in this efficiency calculation and what blocks are not (digital parts of network, laser, temp. controller, ...).

Reply:

Thanks for the suggestion.

In the excepted energy efficiency calculation, the power consumption of the pump laser, the on-chip SOA (take the place of EDFA), and the electronic blocks (including integrated driver, TIA, DAC, ADC) is included. The power consumption of the discrete EDFA, the digital backend and the temperature controllers is excluded in this efficiency calculation. The detailed power consumption estimation procedure of our PPU has been clearly shown in our supplementary information note IX.

Response to Reviewer #2

General comments:

The authors did an admirable job and fully addressed my comments. I have no further suggestions and support the publication of this work.

Reply:

Thanks for this constructive comment and the support for publication of our work.